# HYBRID MINORITY OVERSAMPLING VIA LLM-GENERATED SEEDS AND SMOTE EXPANSION

## ABSTRACT

Class imbalance poses a persistent challenge in machine learning, as classifiers often underperform on the minority class when trained on skewed data. Oversampling is a common solution, with methods such as Synthetic Minority Oversampling Technique (SMOTE) offering efficiency but limited representational power, since they rely solely on existing data points. Recent approaches that employ large language models (LLMs) for oversampling overcome this limitation by generating diverse synthetic samples informed by contextual knowledge. However, LLM-only methods are computationally expensive and often impractical at scale. To bridge this gap, we propose **LLM-SMOTE Hybrid (LSH)**, a method that integrates the strengths of both paradigms. In LSH, an LLM acts as a *Scout* that generates contextually meaningful seed samples for the minority class, while SMOTE serves as the *Surveyor* that efficiently expands these seeds to generate new samples. This design reduces reliance on repeated LLM calls while preserving diversity and scalability. Extensive experiments on 60 imbalanced tabular datasets, across multiple classifiers and resampling strategies, reveal that LSH consistently outperforms SMOTE and LLM in highly imbalanced datasets, demonstrating particular effectiveness in few-shot and zero-shot scenarios where SMOTE fails. Robustness analysis further shows that LSH achieves more stable generalization with lower variance than other methods. Finally, LSH provides a practical trade-off, achieving competitive performance to LLM-based methods at substantially lower computational cost. These findings position LSH as an efficient, robust, and broadly applicable oversampling strategy for imbalanced learning problems.

## 1 INTRODUCTION

Learning from imbalanced datasets is a long-standing challenge in machine learning. In many real-world applications—such as fraud detection (Rubaidi et al. (2022)), medical diagnosis (Salmi et al. (2024)), and rare event prediction (Ribeiro & Moniz (2020))—the distribution of classes is highly skewed, causing standard classifiers to favor the majority class and underperform on the minority class. Oversampling methods, which artificially generate additional instances of the minority class, are widely used to resolve this problem (Mujahid et al. (2024)).

Traditional oversampling techniques, such as the Synthetic Minority Oversampling Technique (SMOTE) (Chawla et al. (2002)), create new synthetic samples by interpolating between existing minority instances. While effective and computationally inexpensive, these methods are fundamentally limited: they can only exploit the information contained in the available data points (Li et al. (2025); Khorshidi & Aickelin (2020)). When minority samples are scarce or absent, they may overfit or fail altogether, as they lack sufficient information to expand the feature space meaningfully.

Recent advances in large language models (LLMs) opened up new possibilities for oversampling. Unlike interpolation-based methods, LLMs can leverage their broad contextual knowledge to generate synthetic data that is not confined to existing data. Early studies show that LLMs can produce plausible and diverse tabular data, often outperforming traditional methods (Kim et al. (2024); Borisov et al. (2022)). However, such LLM-only methods are typically computationally expensive, requiring multiple prompt–generation cycles and significant resources (Nguyen et al. (2025)). This limits their practicality, particularly for large-scale or resource-constrained applications.

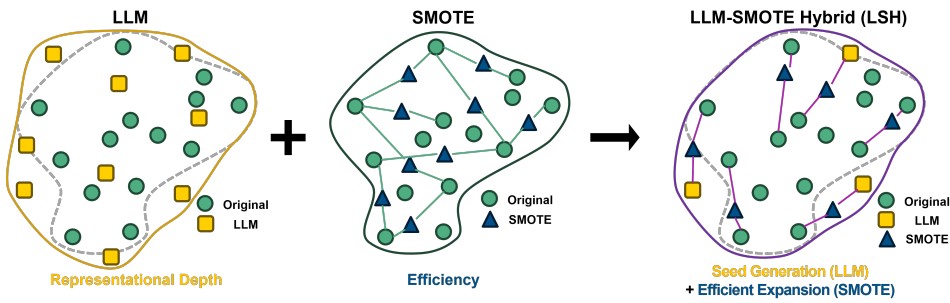

Figure 1: LSH idea. We leverage LLMs' representational power (left) and SMOTE's efficiency (center). In the LSH approach (right), an LLM explores the feature space to generate contextually meaningful seed samples. Then, SMOTE efficiently fills the expanded space through interpolation.

To address this gap, we propose a hybrid strategy that combines the complementary strengths of LLMs and SMOTE, as shown in Figure 1, called **LLM-SMOTE Hybrid (LSH)**: *An LLM acts as a **Scout**, exploring the feature space to generate diverse and contextually meaningful seed samples (yellow) for the minority class. SMOTE then plays the role of the **Surveyor**, efficiently filling the space around minority data points through interpolation (navy).* This Scout–Surveyor approach captures both the representational depth of LLMs and the expansion efficiency of SMOTE.

The central motivation is to evaluate whether this hybrid approach provides consistent advantages, i.e., reasonable performance with efficiency, over the SMOTE and LLM-only methods across a broad range of imbalanced learning scenarios. Specifically, we investigate three key settings: general imbalanced datasets, varying imbalance severity, and extreme few-shot and zero-shot cases. We also assess robustness, defined as the consistency of generalization from training to validation and test performance, and we discuss efficiency trade-offs between LSH and the LLM-only method.

**Novelty and Contributions.** This work introduces a new perspective on oversampling for imbalanced tabular data by leveraging the complementary strengths of LLMs and SMOTE. While traditional oversampling techniques, such as SMOTE, are efficient, their representation capability is constrained by existing data points. In contrast, LLM-based methods can generate diverse synthetic samples, but at a high computational cost. Our method aims to bridge this gap, and the following contributions summarize the novelty and significance of our study:

1. **Hybrid Oversampling Design**: We propose a new method, **LLM-SMOTE Hybrid (LSH)**, that combines the representational power of LLMs with the efficiency of SMOTE. An LLM acts as a Scout to generate contextually meaningful seed samples, while SMOTE serves as the Surveyor to expand these seeds with incomparable efficiency.

2. **Extensive Empirical Evaluation**: We conducted large-scale experiments across 60 imbalanced tabular datasets, ensuring comprehensive and generalizable results. Evaluation with 695 model configurations (139 classifiers and 5 resampling strategies) for each oversampling method and five-fold cross-validation provides a fair comparison.

3. **Performance in Highly Imbalanced and Extreme Settings**: The results show that LSH consistently outperforms SMOTE and LLM in highly imbalanced settings. Moreover, in the extreme scenarios (few- and zero-shot), where SMOTE fails, LSH remains effective and provides slight but meaningful improvements over the LLM-only method.

4. **Robustness Across Datasets**: We introduce an achievement rate analysis to measure the consistency of performance between training and testing. LSH shows lower variance and more stable generalization compared to the SMOTE and LLM-only methods.

5. **Efficiency–Performance Trade-off**: Unlike prior LLM-only approaches that require repeated and expensive model queries, LSH invokes the LLM only for initial seed generation. The subsequent expansion is handled by SMOTE, yielding a practical balance between representational diversity and computational efficiency.

The remainder of this paper is organized as follows: Section 2 reviews related work, Section 3 presents the proposed method, Section 4 describes the experimental setup, Section 5 reports the results, and Section 6 provides a discussion, and Section 7 concludes with future directions.

## 2 RELATED WORK

Research on handling class imbalance has led to a broad spectrum of oversampling strategies, ranging from traditional methods to advanced approaches that leverage LLMs. We provide a brief review of these categories and their limitations in the context of oversampling tabular data.

**Traditional Methods** Oversampling techniques are widely used to address class imbalance in machine learning. The most prominent method, SMOTE, generates synthetic minority samples by interpolating existing ones. Many variants of SMOTE have been proposed, including Borderline-SMOTE (Han et al. (2005)), ADASYN (He et al. (2008)), and SMOTE-ENN (Batista et al. (2004)), which attempt to refine the generation process by focusing on decision boundaries or adaptively weighting instances. Despite their popularity and efficiency, all interpolation-based methods share a common limitation: they entirely rely on the distribution of available minority samples (Li et al. (2025); Khorshidi & Aickelin (2020)). When the number of minority points is scarce, these methods may generate redundant or uninformative data, and in extreme cases, they may not operate at all.

**LLM-based Methods** The emergence of LLMs has opened up a new era for generating tabular data. Recent studies have explored the use of LLMs to oversample tabular datasets by framing the generation process as a text-to-table task (Isomura et al. (2025); Törnqvist et al. (2025); Wang et al. (2024)). These approaches leverage LLMs' contextual understanding and extensive knowledge base to generate diverse, semantically meaningful synthetic samples. Recent works show that LLM-based oversampling can outperform traditional methods, particularly when the minority class is scarce. However, these methods come at a high computational cost. Generating synthetic data with LLMs often requires multiple prompt–response cycles and significant resource consumption, making them impractical in large-scale or resource-constrained environments (Chan et al. (2024)).

**Hybrid Approaches** To the best of our knowledge, prior work has largely treated SMOTE-based and LLM-based oversampling as separate directions. Hybrid approaches that explicitly combine the efficiency of interpolation methods with the representational power of LLMs remain unexplored. Our work fills this gap by proposing the **LLM-SMOTE Hybrid (LSH)** method, which utilizes LLMs to generate a small yet diverse set of seed samples and then employs SMOTE to efficiently expand these seeds. This design aims for availability, robustness, and computational efficiency.

## 3 METHOD

We describe our hybrid method, LSH, in this section. For the interpolation-based component, we use the standard SMOTE because it is simple and widely recognized as the canonical baseline in oversampling. For the LLM component, we employ *GPT-4o-mini* (OpenAI (2024)), chosen for its strong generative ability and practical accessibility (i.e., via an API without significant modifications).

### 3.1 SMOTE OVERSAMPLING

We use SMOTE for our hybrid approach. This choice is motivated by its status as the canonical baseline for oversampling, from which many subsequent methods are derived. SMOTE generates synthetic minority class samples by interpolating existing minority points. For each minority instance, SMOTE selects its k nearest minority neighbors and creates new samples along the line segments connecting the instance to its neighbors. This method is simple, computationally efficient, and widely used. However, SMOTE is fundamentally limited by the number and distribution of existing minority points, and it may not generate meaningful samples in few-shot or zero-shot scenarios. Its simplicity and efficiency align well with our hybrid design: SMOTE provides a scalable expansion mechanism that complements the representational depth introduced by LLMs, making it an ideal partner in the proposed LSH framework. Although SMOTE is solely used in this paper, the hybrid approach is not restricted to SMOTE. Other interpolation methods (e.g., SMOTE variants)

could be substituted in future work, depending on domain needs. Our focus here is to establish the hybrid paradigm, not to exhaustively benchmark all possible interpolators; therefore, we use the representative one, SMOTE.

### 3.2 LLM-Based Oversampling

For the LLM-based method, we use a pipeline that employs *GPT-4o-mini*. We chose this API-based model because it provides a strong balance of capability and accessibility: it is powerful enough to capture complex relationships in tabular data, while also being simpler than complex alternatives such as prior works (Isomura et al. (2025); Törnqvist et al. (2025); Wang et al. (2024)). Although any sufficiently capable LLM could play the same role within the LSH method (i.e., LSH is model-agnostic), our goal is not to benchmark different models but to demonstrate the feasibility of combining an LLM with an interpolation-based method in a hybrid framework; therefore, we do not consider other LLMs in this paper. Details about our pipeline using *GPT-4o-mini* are provided in the Appendix A.2.

### 3.3 Hybrid Oversampling: LSH (LLM + SMOTE)

LSH combines the complementary strengths of LLMs and SMOTE in a two-stage process. First, the LLM generates a small set of contextually meaningful minority seeds, effectively expanding the representational boundary of the minority class. Then, SMOTE efficiently generates additional synthetic samples, filling in the feature space with diverse yet computationally inexpensive samples. This design achieves a balance between representational depth and efficiency. LLMs enable the exploration of data beyond existing limitations, which is particularly valuable in severe-imbalance, few-shot, or zero-shot scenarios. At the same time, SMOTE ensures scalability by handling the bulk of data generation at negligible computational cost. By invoking the LLM only for seed creation and then relying on SMOTE for expansion, LSH avoids the repeated overhead of LLM-only methods while preserving their representational benefits. In this way, the LSH achieves both robustness and efficiency, making it well-suited for practical applications. The actual distribution of sample data across different oversampling methods (Figure 2, PCA used for feature reduction) effectively illustrates the concept already shown in Figure 1. SMOTE fills the space from the original distribution, while LLM expands the original distribution. In LSH, the LLM slightly expands from the Original distribution, and SMOTE fills the gap.

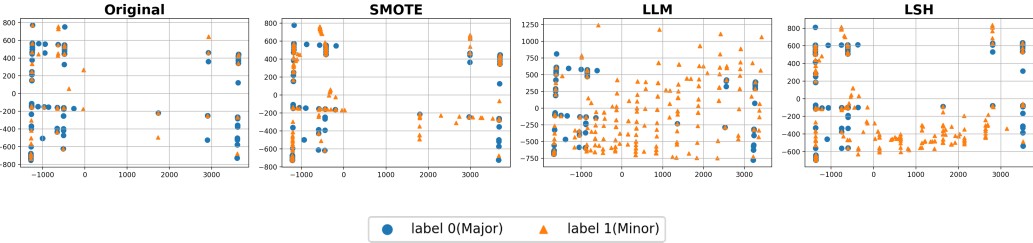

Figure 2: The actual distribution (PCA-based) of the sample data across different oversampling methods illustrates the LSH concept as shown in Figure 1. SMOTE fills the space within the original distribution, whereas LLM expands it. In LSH, the LLM slightly expands from the original distribution, and SMOTE fills the gap.

## 4 Experiments

We present a basic experimental setup and diverse scenarios for evaluating different oversampling methods, followed by research questions. Figure 3 describes the detailed experimental settings. As shown in (a), the goal of the experiment is to compare three oversampling methods (i.e., SMOTE, LLM, and LSH) across 60 imbalanced binary datasets. The final result is the raw score table across all datasets (the green table), which can be used for diverse analyses. The process for getting the final score in one dataset is shown in (b). We employed a five-fold cross-validation protocol on 70% of the data (training), with the remaining 30% held out for testing. Cross-validation identifies the best

model configuration, and the test result is obtained with the best option for each method. F1-score is used as the primary evaluation metric because we are dealing with an imbalanced classification problem. All considered configurations are provided in (c). Logistic Regression (LR), Decision Tree (DT), Support Vector Machine (SVM), k-Nearest Neighbors (kNN), and LightGBM (LGBM) are used as classifiers. A total of 139 hyperparameter combinations within classifiers were considered. Details are provided in the Appendix A.3 (Table 2). Concerning the resampling strategy, for SMOTE and LLM-only methods, the minority class is generated until target majority-to-minority ratios of 1:0.2, 1:0.4, 1:0.6, 1:0.8, and 1:1 are reached. For LSH, the LLM generates initial seed samples to reach the first target ratio (e.g., 1:0.2), and SMOTE scales to the remaining ratios. Determining the optimal initial seed ratio is rarely possible because the appropriate seed ratio can vary across datasets and LLMs. For our motivation, i.e., efficiency, we minimize LLM invocation.

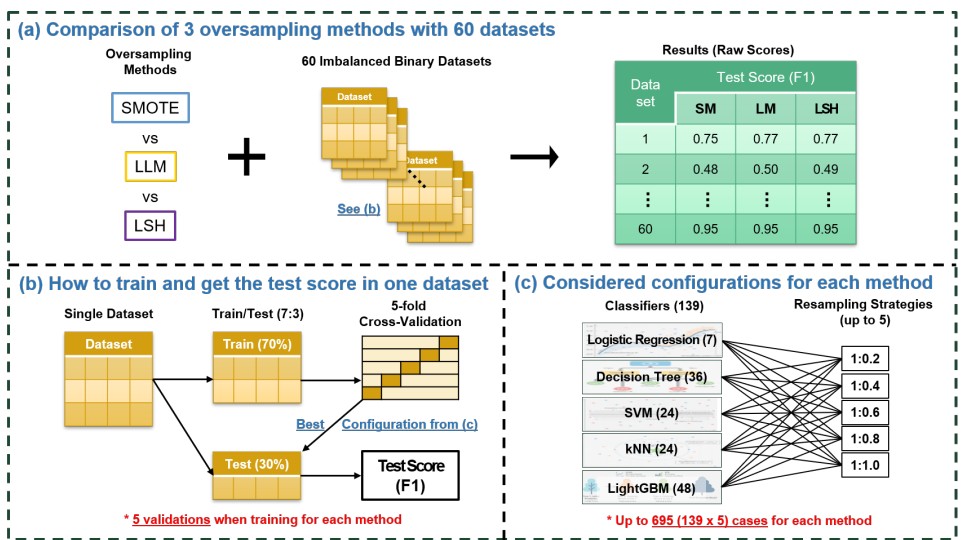

Figure 3: Experimental settings. (a) The goal of the experiment is to compare three oversampling methods across 60 datasets. (b) In one dataset, 5-fold cross-validation is used for training to select the best model configuration, which is used for testing. (c) Up to 695 model configurations are considered during training for each resampling method.

This design enables a fair comparison among different oversampling methods, as each method may have its optimal combination of classifier, hyperparameters, and resampling strategy within each dataset. The five-fold cross-validation also ensures the robustness of the selection, resulting in trustworthy test results. For a detailed analysis, we considered the two different scenarios, 'General Imbalanced Scenario' and 'Extremely Imbalanced Scenario (Few-Shot and Zero-Shot)'.

## 4.1 GENERAL IMBALANCED SCENARIO

To evaluate the effectiveness of LSH under realistic data imbalance, we conducted experiments on 60 benchmark tabular datasets collected by Moniz et al. (Moniz & Cerqueira (2021)) from *OpenML*, which contain varying sample sizes, feature counts, and imbalance ratios (IRs). The IRs range from moderately skewed distributions (∼12:1) to near-balanced settings (∼1.5:1), enabling a comprehensive assessment across different degrees of class rarity. To better observe, we divide the datasets into three equal-sized groups based on their IRs: **More imbalanced/Mid imbalanced/Less imbalanced** groups. The 20 datasets with higher IRs are assigned to the More Imbalanced group, followed by the next groups. Details are provided in the Appendix A.5 (Table 6).

## 4.2 EXTREMELY IMBALANCED SCENARIO (FEW-SHOT AND ZERO-SHOT)

To further investigate the robustness of LSH, we considered extreme imbalance conditions, where the minority class constitutes only a tiny fraction of the data. Specifically, we modified four selected datasets to simulate scenarios with class ratios (majority-to-minority) of 1:0.01, and 1:0.00 in the

training sets, while moving the removed minority samples from the training sets to the validation and test sets. These harsh conditions represent realistic challenges in applications such as fraud detection, medical diagnosis, and anomaly detection. Few-shot experiments involved keeping a minimal subset of minority samples in training, while zero-shot experiments removed all minority training samples. In both cases, models were evaluated on test sets containing actual minority instances.

### 4.3 RESEARCH QUESTIONS

To systematically evaluate the effectiveness of the proposed method, we formulate four research questions. They are designed to capture the central themes of our work: sensitivity to the severity of imbalance, behavior in extreme settings, robustness across datasets, and efficiency trade-offs. Each research question is directly linked to a corresponding set of experiments and analyses.

- RQ1. Does the relative advantage of LSH increase with imbalance severity? (5.1)
- RQ2. Can LSH remain effective under extreme scenarios where SMOTE fails? (5.2)
- RQ3. How robust are SMOTE, LLM, and LSH across datasets? (5.3)
- RQ4. Does LSH provide a practical efficiency compared to LLM-only methods? (5.4)

## 5 RESULTS AND ANALYSIS

In this section, we present the performance evaluation of the three oversampling methods—SMOTE (SM), LLM-based (LM), and the proposed LSH (LS)—across the 60 datasets. We analyze overall performance, performance by imbalance severity, performance in few-shot and zero-shot scenarios, robustness, and efficiency. In addition to simple score comparisons and data visualization, statistical analysis methods were used, including correlation coefficients and the Bayesian Sign Test (BST) (Benavoli et al. (2017)). BST calculates the probability that one method outperforms another across datasets. BST provides win/draw/lose probabilities for probabilistic comparisons, enabling robust insights across heterogeneous datasets.

Before we address the research questions, we observed performance across the three oversampling methods on all 60 datasets. The detailed analysis is provided in Appendix A.4 (Table 3). The results show that none of the methods universally dominates. This is the expected result because performance varied with dataset characteristics, classifiers, and resampling strategies. This aligns with prior findings in the imbalance literature, where no oversampling method is universally optimal (Moniz & Monteiro (2021)).

### 5.1 PERFORMANCE BY IMBALANCE SEVERITY (RQ1)

To better understand how oversampling methods behave under different Imbalance Ratios (IRs), we divided the 60 datasets into three groups of equal size: **more imbalanced**, **mid imbalanced**, and **less imbalanced**. Details about three groups are provided in the Appendix A.5 (Table 6). This grouping allows us to observe trends in performance as class imbalance becomes more or less severe.

Figure 4a shows the comparisons (BST) between methods in the three groups. In the comparison between LSH and LLM (center) and between LSH and SMOTE (right), the draw probabilities decrease (LS=LM and LS=SM, beige, $89.5\% \rightarrow 45.5\%$ and $11.8\%$), and the win probabilities increase (LS>LM and LS>SM, blue, $5.3\% \rightarrow 45.7\%$ and $88.1\%$) as the dataset imbalance increases. No explicit patterns are observed in the comparison between LLM and SMOTE (left). The results illustrate that the hybrid approach primarily benefits datasets with higher imbalance. The detailed results, including score margins, are provided in Appendix A.4 (Table 4).

Figure 4b further highlights these patterns. The score margins with LSH and IRs exhibit a positive correlation with imbalance severity, both over LLM (center, Pearson = 0.2587, $p < 0.05$) and especially over SMOTE (right, Pearson = 0.5588, $p < 0.01$; Spearman = 0.3723, $p < 0.01$). No correlation between the score margin of LLM over SMOTE and IR (left). These findings indicate that the relative advantage of LSH increases as datasets become more imbalanced, aligning with the intuition that using LLM-generated seeds is particularly beneficial when minority data are scarce, enabling SMOTE to interpolate effectively from a more representative foundation.

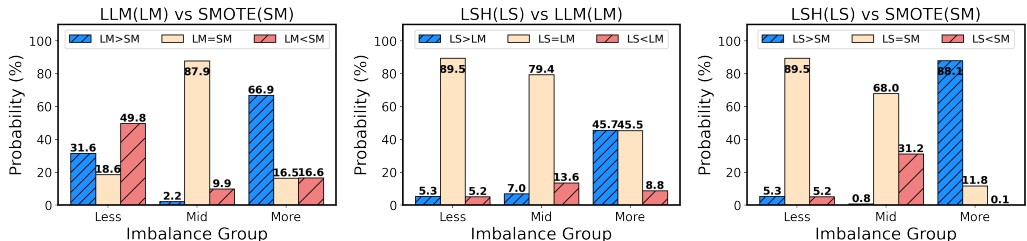

(a) Comparisons (BST) between methods. For LSH vs LLM (center) and LSH vs SMOTE (right), the draw probabilities decrease (LS=LM and LS=SM, 89.5% → 45.5% and 11.8%), and the win probabilities increase (LS>LM and LS>SM, 5.3% → 45.7% and 88.1%) as the dataset imbalance increases.

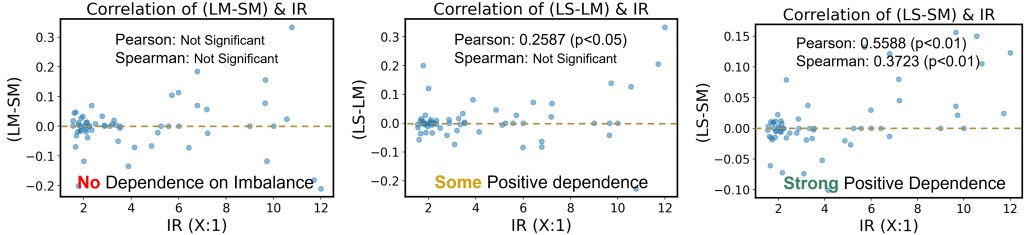

(b) Correlation between score margins and IRs. The score margins with LSH and IRs exhibit a positive correlation with imbalance severity, both over LLM (center, Pearson = 0.2587, p < 0.05) and especially over SMOTE (right, Pearson = 0.5588, p < 0.01; Spearman = 0.3723, p < 0.01).

Figure 4: [RQ1] Performance Comparison by Imbalance Severity

## 5.2 FEW-SHOT AND ZERO-SHOT EXPERIMENTS (RQ2)

We evaluated performance under extreme few-shot and zero-shot conditions. To avoid biased results, we selected four datasets with diverse characteristics (e.g., sample sizes and the number of numerical and categorical features), as shown in Table 1. Across four selected datasets, SMOTE failed in the 1:0.01 and 1:0.00 cases due to insufficient minority points. In these scenarios, both LLM and LSH remained functional, with LSH consistently showing slight improvements over LLM. For example, in the few-shot setting, LSH achieved higher F1 scores than LLM across most datasets, and in the zero-shot setting, LSH either matched or slightly exceeded LLM's performance. These results highlight the practical advantage of combining LLM generation with SMOTE: LSH inherits LLM's capability to produce meaningful synthetic data in extremely scarce scenarios while benefiting from SMOTE's efficient scaling. The visualized data distributions (PCA-based) under these scenarios are provided in the Appendix A.6 (Figure 8).

Overall, the results demonstrate that while no single oversampling method strictly dominates across all datasets, LSH consistently provides good representation power (performance) and computational efficiency (limited LLM usage). Its advantages are most pronounced in extreme few-shot and zero-shot settings, supporting the rationale behind the Scout and Surveyor metaphor: the LLM explores the feature space deeply, and SMOTE efficiently expands this exploration.

Table 1: [RQ2] Results in Few-shot and Zero-shot scenarios where SMOTE is not applicable.

| Dataset # | Few-shot (1:0.01) | | | | Zero-shot (1:0.00) | | | |
|---|---|---|---|---|---|---|---|---|
| (S/N/C)* | ORG | SM | LM | LS | ORG | SM | LM | LS |
| 8 (335/1/2) | 0.2703 | N/A | 0.6970 | **0.7246** | N/A | N/A | **0.4342** | **0.4342** |
| 58 (1,066/0/7) | 0.0161 | N/A | 0.5477 | **0.5726** | N/A | N/A | 0.1453 | **0.3599** |
| 14 (365/3/2) | 0.0909 | N/A | 0.2837 | **0.2857** | N/A | N/A | **0.3220** | 0.3182 |
| 44 (748/4/0) | 0.0488 | N/A | **0.6053** | 0.5819 | N/A | N/A | 0.5540 | **0.5674** |

\* S: # Samples / N: # Numerical Features / C: # Categorical Features.

## 5.3 ROBUSTNESS (RQ3)

To assess robustness, we measured the Achievement Rate (AR), defined as the ratio of testing performance to validation performance, across 60 datasets. This metric captures how well each method generalizes beyond validation, and the results are shown in Fig 5. On average, SMOTE (left) had the lowest robustness with a mean AR (0.9146) and the highest standard deviation (0.1888). LLM (center) improved both the mean (0.9406) and standard deviation (0.1739), but variability remained noticeable. LSH (right) achieved nearly the same mean robustness (0.9380) as LLM but with the lowest standard deviation (0.1303), making it the most stable of the three.

Beyond mean and standard deviation, we examined the correlation between AR and IR. For SMOTE, there is a strong negative dependence: robustness decreases sharply with increasing imbalance, Pearson correlation: -0.5128 (p < 0.01); Spearman correlation: -0.3009 (p < 0.05), confirming its fragility under severe imbalance. For LLM, there is some negative dependence, as robustness also declines but less consistently, Pearson correlation: –0.4199 (p < 0.01); Spearman correlation: non-significant, indicating partial resilience. For LSH, there is a moderate negative dependence, where robustness decreases modestly but consistently, as supported by both the Pearson correlation (-0.3244, p < 0.05) and the Spearman correlation (-0.3360, p < 0.01). LSH thus exhibits greater stability than both SMOTE and LLM, although it is not completely insensitive to imbalance.

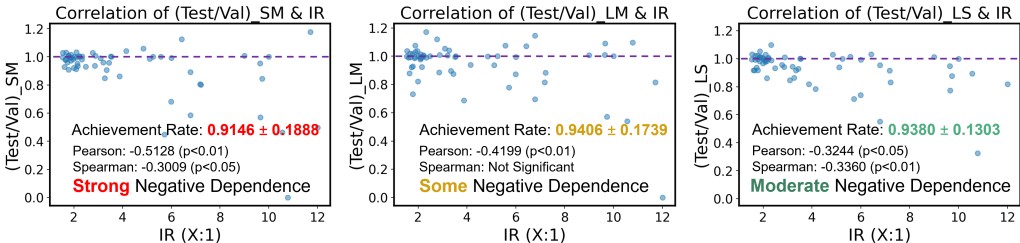

Figure 5: [RQ4] Robustness analysis with AR. LSH (right) achieved nearly the same mean (0.9380) as LLM but with the lowest standard deviation (0.1303). Regarding the correlation between AR and IR, LSH shows a moderate negative dependence; Pearson: -0.3244 (p < 0.05); Spearman: -0.3360 (p < 0.01). LSH thus exhibits greater robustness and stability under imbalance than others.

## 5.4 EFFICIENCY (RQ4)

While LLM-based oversampling can generate contextually meaningful samples, it is prohibitively slow because the LLM must be invoked repeatedly for every resampling ratio and dataset. In contrast, LSH requires the LLM only when generating seed samples, after which SMOTE handles subsequent resampling at negligible computational cost. This design ensures that LSH avoids its runtime bottlenecks. To quantify this, we measured runtime on two datasets: 5- and 37-feature datasets. Oversampling was performed at multiple resampling ratios. Results shown in Figure 6 provide a clear observation.

For the 5-feature dataset, LLM-only required between 65 and 230 seconds per resampling run, with runtime increasing as the resampling ratio grew. By contrast, LSH took only about 65 seconds in total, since the LLM was called once at the initial ratio (0.2), and SMOTE performed subsequent expansions at higher ratios in milliseconds. A similar pattern emerged for the 37-feature dataset, where LLM runtimes ranged from 129 to 563 seconds, while LSH stabilized around 129 seconds regardless of the resampling strategy. Flat parts are observed because the number of generated samples varies when the LLM is called, depending on factors such as feature type and feature count.

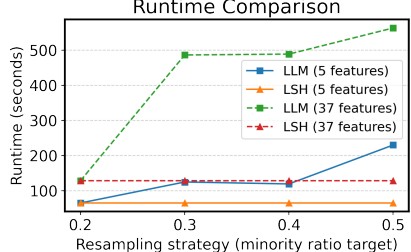

Figure 6: [RQ5] Runtime comparison. LSH (red, yellow) shows no increase with higher resampling rates.

These results demonstrate two key points. First, LLM-only oversampling scales poorly with both feature dimensionality and resampling strategy, making it impractical for large-scale use. Second, LSH maintains constant runtime after the initial seed generation, improving efficiency. This confirms that the hybrid design of LSH not only improves representational capacity in challenging imbalance settings but also makes oversampling computationally practical.

# 6 DISCUSSION

**For the RQ1**, the evidence strongly supports this: **LSH shows clear advantages in the more imbalanced group**, with higher win probability over both SMOTE and LLM, and the strongest positive correlation between performance margin and imbalance ratio. This suggests that the hybrid design, using the LLM as a Scout to establish diverse minority landmarks before applying SMOTE as a Surveyor, effectively addresses the challenges posed by imbalance.

**For the RQ2**, the results confirm that **LSH, like LLM, remains functional in the extreme conditions, but LSH consistently provides slight but meaningful improvements over LLM**. This demonstrates the value of combining contextual exploration by LLM with efficient expansion by SMOTE, even when the initial data is severely limited or absent.

**For the RQ3**, the achievement rate analysis reveals that SMOTE is fragile, exhibiting a strong negative dependence between robustness and the imbalance ratio. LLM mitigates this fragility, albeit with limited resilience. **LSH achieves the best stability, with a moderate negative dependence that is weaker and more consistent than SMOTE or LLM**.

**For the RQ4**, unlike LLM-only oversampling, which requires a full model invocation for every resampling strategy and can take minutes per run, **LSH minimizes LLM usage by generating seed samples only once and relying on SMOTE for subsequent expansion**. The results highlight that LSH transforms oversampling into a constant-time process after the initial seed generation. Put differently, the Scout (LLM) identifies the landmarks at a one-time cost, and the Surveyor (SMOTE) expands the map at negligible additional expense.

Taken together, these findings highlight the strengths of our proposed method. Although it does not dominate others, it excels where traditional oversampling struggles most: in severe imbalance, few-shot, and zero-shot scenarios. It provides favorable robustness and efficiency. As such, LSH represents a practical step forward for oversampling in imbalanced tabular learning, particularly when scalability and stability are required.

# 7 CONCLUSION

This paper introduced LSH, a hybrid oversampling method that leverages the complementary strengths of LLMs and SMOTE for imbalanced tabular data. Across 60 imbalanced binary datasets, we demonstrated that LSH outperforms SMOTE and LLM in severe imbalance, few-shot, and zero-shot settings where traditional methods fail. Robustness analysis further showed that LSH is more stable than either baseline, exhibiting lower variance in achievement rates and only moderate sensitivity to imbalance. Notably, the design of LSH also ensures computational efficiency by limiting LLM usage to a small set of seed samples, with SMOTE handling scalable expansion.

While these findings highlight the strengths of LSH, several considerations for future work remain. First, the exact usage of the LLM, specifically, how many seed samples should be generated for optimal performance, remains unclear. Although a fixed number cannot be specified, we can examine whether it depends on factors such as the data domain or data characteristics. Second, we focused exclusively on binary classification, and extending LSH to multi-class or regression tasks represents a natural next step. Future research can build upon this foundation by clarifying the optimal role of LLM-generated seeds and extending the method to more complex learning scenarios. For reproducibility, the code for experiments in this work is available at https://anonymous.4open.science/r/LSH-D711/README.md.

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

## A  APPENDIX

### A.1  USE OF LARGE LANGUAGE MODELS

For this paper, we utilized large language models (LLMs) as supporting tools, including grammar correction, writing polish, converting tables into LaTeX format, and searching reference papers.

### A.2  OUR LLM-BASED OVERSAMPLING METHOD

# LLM-Based Tabular Data Oversampler

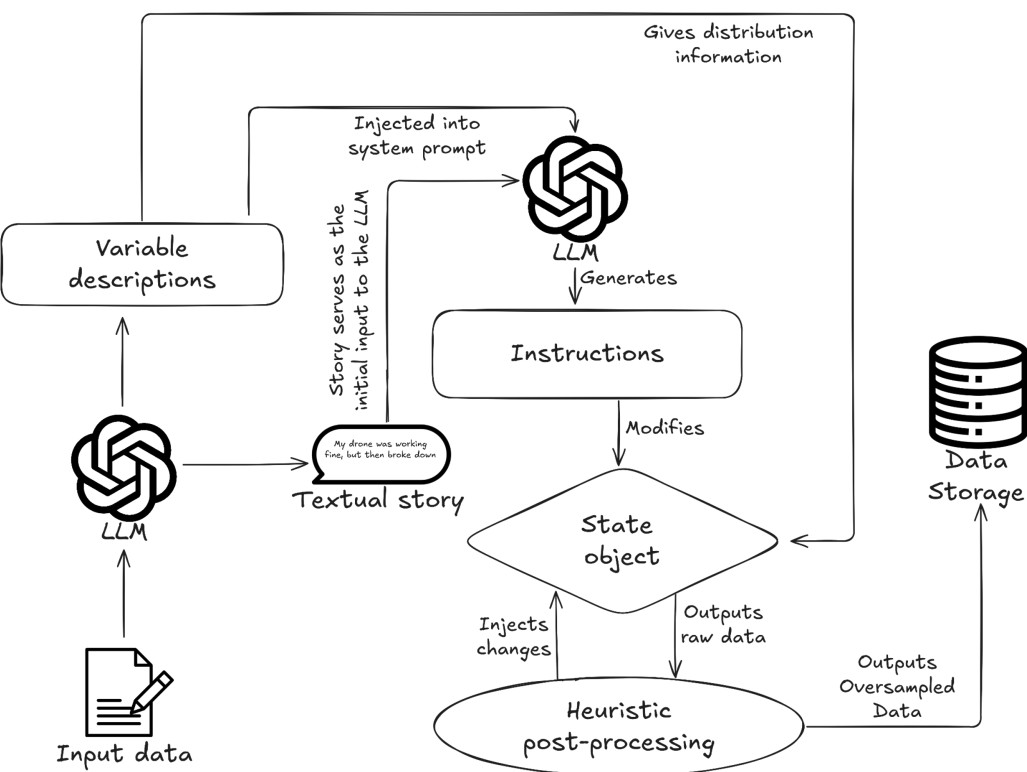

Figure 7: The concept of our LLM-based oversampling method,

The oversampling pipeline proceeds in several stages as described in Figure 7. First, we draw a sample from the dataset we are trying to oversample and prompt the LLM to generate a rich, textual

description of the dataset. Additionally, we use constrained generation to extract the descriptions and the potential distributions of the variables in the dataset based on the provided sample.

Next, the LLM is prompted to generate a sequence of instructions for an abstract machine. In this procedure, the instruction set is provided to the LLM, and constrained generation is used to ensure that the LLM generates syntactically correct code. The instruction set includes instructions that allow for setting a variable to a value, changing a variable by a delta, and outputting the current state of the machine. On output, the machine's outputs are jittered by a small amount, utilizing the distributions extracted by the model. The outputs of the abstract machine constitute the oversampled data. We then rerun this pipeline until we obtain the desired number of synthetic samples.

This method maximizes the use of code-generation capabilities of the current generation of LLMs, while avoiding directly generating large amounts of numerical data. At the same time, this method does not require access to model logits or any other model outputs other than the generated tokens; this allows using this method with both open-source models, and LLMs that are only available as limited APIs.

Furthermore, we provide the LLM prompts we used on each step of the oversampling pipeline described in Figure 7:

**[Data description prompt]:**

```
You're an expert data analyst at a large company. Your boss has
    asked you to analyze a dataset and provide a report on your
    findings.

You'll be provided with a dataset in CSV format with a header. You
    'll need to write a report that provides insights into the
    data. Focus on the key points that will help your boss make
    informed decisions. Put a scpecial emphasis on the
    relationships between the different variables in the dataset,
    the trends that you observe.

Make sure to try to extract the general meaning of the data, and
    not just the raw numbers. Your boss is looking for a high-
    level summary of the data, not a detailed analysis.

Your report should be clear, concise, and easy to understand. It
    should be written in plain English, and should not contain any
     technical jargon. At least 500 words.
```

**[Variable description prompt]:**

```
You're an expert data analyst at a large company. Your boss has
    asked you to analyze a dataset and provide a report on your
    findings.

You'll be provided with a dataset in CSV format with a header.
    Your task is to extract the names, descriptions and possible
    distributions of the variables in the dataset. Please note
    that categorical and string variables use the "none"
    distribution.

You should provide your answer as a machine-readable JSON object.
    The object should follow this schema:

$VARIABLE-EXTRACTION-JSON-SCHEMA
```

* Where `$VARIABLE-EXTRACTION-JSON-SCHEMA` is a dynamically generated JSON schema of the desired output format (described in code as a Pydantic model) that is also enforced by constrained generation.

**[Data generation prompt]:**

Your task is to control a synthetic data generator that creates
    synthetic data conforming to a report given to you.

The generator is a state machine that has an internal state which
    can be sent to the output, or that can be changed. You can use
     functions to control the state machine. Each output of the
    machine describes a single data point.

You will be given a report by the user that the synthetic data
    should follow as closely as possible. Ensure that you only use
     the variables provided to you by the user. Make sure that the
     synthetic data you generate is consistent with the report.
    Make sure to use the EXACT variable names provided in the
    report.

You should just output the sequence of operations without any text
     in natural language.

The change_by operation cannot be used on variables with units "
    true or false" or string variables.

Your output should be a single JSON object following the schema
    below:
$DATA−GENERATION−JSON−SCHEMA

Do not use the 'change_by' operation on variables with units "true
     or false", string or categorical variables.

The sequence of operations should output at least $N−OUT data
    points.

An example of the data that you should try to generate is:
'''
$DATA−EXAMPLE
'''

(this is just an example of the data format, do not try to
    replicate the data itself, only the overall format)

The sequence of operations should be consistent with the report
    and follow the schema above.

You must generate a sequence of instructions that will output at
    least $N−OUT data points.

You have access to the following variables (their exact names
    between | symbols):
$VARIABLE−DESCRIPTION−TEXT

* Where $DATA-GENERATION-JSON-SCHEMA is a dynamically generated JSON schema describing the output format (also described as a Pydantic model, and enforced via constrained generation); $N-OUT is the desired count of generated samples (provided by the user); $DATA−EXAMPLE is a small sample of the pipeline input data; $VARIABLE-DESCRIPTION-TEXT is a dynamically generated textual version of the variable descriptions (names, units, minimum and maximum values, distributions) as extracted by the model.

## A.3 CLASSIFIERS

Table 2 shows all hyperparameter settings for the five classifiers.

Table 2: the 139 hyperparameter settings of classifiers.

| Classifier | Combination | Hyperparameters |
|---|---|---|
| LR | 7 | C:[0.001, 0.01, 0.1, 1, 10, 100, 1000] |
| DT | 4X3X3 =36 | max depth:[10,20,30,40]
min samples split:[2,4,6]
min samples leaf:[1,2,3] |
| SVM | 3X2X4 =24 | C:[0.1, 1, 10]
kernel:[rbf, sigmoid]
gamma:[scale, auto, 0.1, 1] |
| kNN | 4X2X3 =24 | n neighbors:[3, 5, 7, 9]
p:[1, 2]
metric:[euclidean, manhattan, minkowski] |
| LGBM | 3X2X2X4 =48 | boosting type : [gbdt, dart, goss]
max depth : [10,20]
learning rate : [0.01,0.05]
n estimators: [50, 100, 150, 200] |

## A.4 EXPERIMENTAL RESULTS

**Overall Performance.** We compare the three oversampling methods across all 60 imbalanced datasets. **The results show that none of the methods universally dominates.**

In Table 3, the simple paired comparisons of one method against another are tied mostly. The average performance margins indicate that LLM (LM) and SMOTE (SM) yield nearly identical results, while LSH (LS) provides minor improvements. Specifically, the average margin is –0.0002 for LLM over SMOTE, +0.0114 for LSH over LM, and +0.0111 for LSH over SMOTE. The raw scores are provided in Table 5. BST outcomes confirm these observations. For LLM versus SMOTE (LM vs SM), the results are balanced (30.4% win, 39.5% draw, 30.1% lose), indicating no clear superiority. Comparisons involving LSH show that draws dominate: against LLM (8.1% win, 88.4% draw, 3.5% lose) and against SMOTE (18.2% win, 79.3% draw, 2.5% lose). These findings suggest that while LSH offers a slight edge overall, all three methods are competitive.

Table 3: Performance across 60 datasets.

| Comparison | 60 results | Average Margin | BST (prob.) |
|---|---|---|---|
| LM vs SM | LM>SM: 26
LM=SM: 12
LM<SM: 22 | LM−SM: -0.0002 | LM>SM: 30.4%
**LM=SM: 39.5%**
LM<SM: 30.1% |
| LS vs LM | LS>LM: 23
LS=LM: 14
LS<LM: 23 | LS−LM: 0.0114 | LS>LM: 8.1%
**LS=LM: 88.4%**
LS<LM: 3.5% |
| LS vs SM | LS>SM: 24
LS=SM: 13
LS<SM: 23 | LS−SM: 0.0111 | LS>SM: 18.2%
**LS=SM: 79.3%**
LS<SM: 2.5% |

**Effectiveness in three imbalanced groups.** In the more imbalanced group, LLM (LM) outperformed SMOTE (SM), with a BST win probability of 66.9%, reflecting LLM's ability to leverage contextual knowledge when few minority samples are available. LSH (LS) demonstrated further improvements, showing the highest probability of outperforming SMOTE (88.1%) and a clear margin (0.0492). For mid-imbalanced datasets, performance differences were minimal, and BST indicated a strong tendency toward draws between methods, confirming that the choice of oversampling method has less impact when the imbalance is moderate. In less imbalanced datasets, LSH maintained a slight advantage, although the differences were minor, illustrating that the hybrid approach primarily benefits datasets with higher imbalance.

Table 4: [RQ2] Performance in three groups by imbalance.

| Comparison | Group | Average Margin | BST (prob.) |
|---|---|---|---|
| | | LM−SM: | LM>SM: / LM=SM: / LM<SM: |
| LM vs SM | More Imb. | **0.0248** | **66.9%** / 16.5% / 16.6% |
| | Mid Imb. | -0.0108 | 2.2% / **87.9%** / 9.9% |
| | Less Imb. | -0.0147 | 31.6% / 18.6% / **49.8%** |
| | | LS−LM: | LS>LM: / LS=LM: / LS<LM: |
| LS vs LM | More Imb. | **0.0245** | **45.7%** / 45.5% / 8.8% |
| | Mid Imb. | -0.0017 | 7.0% / **79.4%** / 13.6% |
| | Less Imb. | 0.0115 | 5.3% / **89.5%** / 5.2% |
| | | LS−SM: | LS>SM: / LS=SM: / LS<SM: |
| LS vs SM | More Imb | **0.0492** | **88.1%** / 11.8% / 0.1% |
| | Mid Imb. | -0.0124 | 0.8% / **68.0%** / 31.2% |
| | Less Imb. | -0.0032 | 5.3% / **89.5%** / 5.2% |

Table 5 shows the raw scores of the three methods.

Table 5: Validation score (_v), test score (_t), achievement rate (_a) of all methods.

| DATA | SM_v | SM_t | LM_v | LM_t | LS_v | LS_t | SM_a | LM_a | LS_a |
|---|---|---|---|---|---|---|---|---|---|
| 1 | 0.4230 | 0.2105 | 0.3123 | 0.0000 | 0.4078 | 0.3333 | 0.4976 | 0.0000 | 0.8173 |
| 2 | 0.4370 | 0.5143 | 0.4085 | 0.3333 | 0.4222 | 0.5385 | 1.1769 | 0.8159 | 1.2755 |
| 3 | 0.2419 | 0.0000 | 0.3038 | 0.3333 | 0.3230 | 0.1053 | 0.0000 | 1.0971 | 0.3260 |
| 4 | 0.4474 | 0.2069 | 0.4282 | 0.2308 | 0.4000 | 0.3571 | 0.4624 | 0.5390 | 0.8928 |
| 5 | 1.0000 | 1.0000 | 1.0000 | 1.0000 | 1.0000 | 1.0000 | 1.0000 | 1.0000 | 1.0000 |
| 6 | 0.4481 | 0.3784 | 0.4563 | 0.2609 | 0.4561 | 0.4000 | 0.8445 | 0.5718 | 0.8770 |
| 7 | 0.3761 | 0.2143 | 0.3400 | 0.3704 | 0.4796 | 0.3704 | 0.5698 | 1.0894 | 0.7723 |
| 8 | 0.6894 | 0.6575 | 0.7286 | 0.7353 | 0.7325 | 0.6933 | 0.9537 | 1.0092 | 0.9465 |
| 9 | 0.9929 | 1.0000 | 0.9927 | 1.0000 | 0.9891 | 1.0000 | 1.0072 | 1.0074 | 1.0110 |
| 10 | 0.3952 | 0.3167 | 0.3317 | 0.2933 | 0.4361 | 0.3621 | 0.8014 | 0.8842 | 0.8303 |
| 11 | 0.7095 | 0.5714 | 0.7713 | 0.6286 | 0.7117 | 0.6512 | 0.8054 | 0.8150 | 0.9150 |
| 12 | 0.6904 | 0.6154 | 0.6982 | 0.8000 | 0.7726 | 0.7368 | 0.8914 | 1.1458 | 0.9537 |
| 13 | 0.4496 | 0.2632 | 0.4787 | 0.3333 | 0.4536 | 0.2500 | 0.5854 | 0.6963 | 0.5511 |
| 14 | 0.7553 | 0.8500 | 0.7255 | 0.7778 | 0.8249 | 0.8500 | 1.1254 | 1.0721 | 1.0304 |
| 15 | 0.9933 | 0.9848 | 0.9911 | 0.9848 | 0.9911 | 0.9848 | 0.9914 | 0.9936 | 0.9936 |
| 16 | 0.4978 | 0.3396 | 0.5190 | 0.4528 | 0.4989 | 0.3692 | 0.6822 | 0.8724 | 0.7400 |
| 17 | 0.3247 | 0.1455 | 0.3211 | 0.2500 | 0.3897 | 0.2778 | 0.4481 | 0.7786 | 0.7129 |
| 18 | 1.0000 | 1.0000 | 0.9295 | 1.0000 | 0.9867 | 1.0000 | 1.0000 | 1.0758 | 1.0135 |
| 19 | 1.0000 | 1.0000 | 1.0000 | 1.0000 | 1.0000 | 1.0000 | 1.0000 | 1.0000 | 1.0000 |
| 20 | 0.9023 | 0.8889 | 0.9264 | 0.8679 | 0.9192 | 0.8621 | 0.9851 | 0.9369 | 0.9379 |
| 21 | 0.4457 | 0.4715 | 0.4057 | 0.4054 | 0.4383 | 0.4511 | 1.0579 | 0.9993 | 1.0292 |
| 22 | 0.3384 | 0.3523 | 0.2998 | 0.2816 | 0.3212 | 0.2517 | 1.0411 | 0.9393 | 0.7836 |
| 23 | 0.6580 | 0.5672 | 0.6309 | 0.4333 | 0.6294 | 0.5152 | 0.8620 | 0.6868 | 0.8186 |
| 24 | 1.0000 | 1.0000 | 1.0000 | 1.0000 | 1.0000 | 1.0000 | 1.0000 | 1.0000 | 1.0000 |
| 25 | 0.8825 | 0.8846 | 0.8847 | 0.8790 | 0.8806 | 0.8820 | 1.0024 | 0.9936 | 1.0016 |
| 26 | 0.8527 | 0.8163 | 0.8746 | 0.7660 | 0.8619 | 0.8000 | 0.9573 | 0.8758 | 0.9282 |
| 27 | 0.7235 | 0.6560 | 0.7369 | 0.6606 | 0.7157 | 0.6560 | 0.9067 | 0.8965 | 0.9166 |
| 28 | 0.5237 | 0.4444 | 0.4462 | 0.4651 | 0.5574 | 0.4815 | 0.8486 | 1.0424 | 0.8638 |
| 29 | 0.5351 | 0.5138 | 0.5186 | 0.5210 | 0.5250 | 0.4950 | 0.9602 | 1.0046 | 0.9429 |
| 30 | 0.6111 | 0.6047 | 0.5740 | 0.6047 | 0.6269 | 0.5306 | 0.9895 | 1.0535 | 0.8464 |
| 31 | 0.9705 | 0.9771 | 0.9672 | 0.9771 | 0.9671 | 0.9771 | 1.0068 | 1.0102 | 1.0103 |
| 32 | 0.4723 | 0.4426 | 0.5034 | 0.4459 | 0.4864 | 0.4578 | 0.9371 | 0.8858 | 0.9412 |
| 33 | 0.5024 | 0.5172 | 0.4922 | 0.5517 | 0.5338 | 0.5098 | 1.0295 | 1.1209 | 0.9550 |
| 34 | 0.5731 | 0.5333 | 0.5409 | 0.5417 | 0.5638 | 0.5255 | 0.9306 | 1.0015 | 0.9321 |
| 35 | 0.9945 | 0.9968 | 0.9931 | 0.9842 | 0.9918 | 0.9811 | 1.0023 | 0.9910 | 0.9892 |
| 36 | 0.5493 | 0.5393 | 0.5195 | 0.6087 | 0.5629 | 0.6180 | 0.9818 | 1.1717 | 1.0979 |
| 37 | 1.0000 | 1.0000 | 1.0000 | 1.0000 | 1.0000 | 1.0000 | 1.0000 | 1.0000 | 1.0000 |
| 38 | 0.4749 | 0.4769 | 0.4733 | 0.4769 | 0.4733 | 0.4769 | 1.0042 | 1.0076 | 1.0076 |
| 39 | 0.6810 | 0.6370 | 0.6480 | 0.5985 | 0.6558 | 0.5649 | 0.9354 | 0.9236 | 0.8614 |
| 40 | 0.8721 | 0.8293 | 0.8541 | 0.8437 | 0.8519 | 0.8372 | 0.9509 | 0.9878 | 0.9827 |
| 41 | 0.7750 | 0.8000 | 0.7715 | 0.7812 | 0.7715 | 0.7937 | 1.0323 | 1.0126 | 1.0288 |
| 42 | 0.6139 | 0.5600 | 0.5853 | 0.5722 | 0.5853 | 0.5722 | 0.9122 | 0.9776 | 0.9776 |
| 43 | 0.9917 | 0.9859 | 0.9877 | 0.9765 | 0.9917 | 0.9813 | 0.9942 | 0.9887 | 0.9895 |
| 44 | 1.0000 | 1.0000 | 1.0000 | 1.0000 | 1.0000 | 1.0000 | 1.0000 | 1.0000 | 1.0000 |
| 45 | 0.5261 | 0.4848 | 0.4470 | 0.3673 | 0.5049 | 0.4878 | 0.9215 | 0.8217 | 0.9661 |
| 46 | 0.7947 | 0.7814 | 0.7856 | 0.8148 | 0.7972 | 0.8036 | 0.9833 | 1.0372 | 1.0080 |
| 47 | 0.5993 | 0.5827 | 0.5682 | 0.6105 | 0.5738 | 0.5781 | 0.9723 | 1.0744 | 1.0075 |
| 48 | 0.9848 | 1.0000 | 0.9709 | 1.0000 | 0.9801 | 0.9950 | 1.0154 | 1.0300 | 1.0152 |
| 49 | 0.6774 | 0.6739 | 0.6622 | 0.6582 | 0.6512 | 0.6854 | 0.9948 | 0.9940 | 1.0525 |
| 50 | 0.9705 | 0.9655 | 0.9642 | 0.9510 | 0.9675 | 0.9510 | 0.9948 | 0.9863 | 0.9829 |
| 51 | 0.9472 | 0.8611 | 0.9411 | 0.8732 | 0.9472 | 0.8696 | 0.9091 | 0.9279 | 0.9181 |
| 52 | 0.6773 | 0.6835 | 0.6604 | 0.4828 | 0.6854 | 0.6835 | 1.0092 | 0.7311 | 0.9972 |
| 53 | 0.9019 | 0.9065 | 0.8967 | 0.9108 | 0.8983 | 0.9167 | 1.0051 | 1.0157 | 1.0205 |
| 54 | 0.9241 | 0.8871 | 0.9417 | 0.8689 | 0.9271 | 0.8689 | 0.9600 | 0.9227 | 0.9372 |
| 55 | 0.9474 | 0.9767 | 0.9522 | 0.9457 | 0.9396 | 0.9612 | 1.0309 | 0.9932 | 1.0230 |
| 56 | 0.9391 | 0.9195 | 0.9474 | 0.9663 | 0.9363 | 0.9302 | 0.9791 | 1.0199 | 0.9935 |
| 57 | 0.7070 | 0.6923 | 0.7084 | 0.6234 | 0.6891 | 0.6316 | 0.9792 | 0.8800 | 0.9166 |
| 58 | 0.7844 | 0.7273 | 0.7793 | 0.7719 | 0.7778 | 0.7143 | 0.9272 | 0.9905 | 0.9184 |
| 59 | 0.8054 | 0.8288 | 0.8164 | 0.8475 | 0.8058 | 0.8319 | 1.0291 | 1.0381 | 1.0324 |
| 60 | 0.8660 | 0.8625 | 0.8641 | 0.8638 | 0.8647 | 0.8595 | 0.9960 | 0.9997 | 0.9940 |

## A.5 DATASET DESCRIPTION

Table 6 describes 60 imbalanced binary tabular datasets.

Table 6: Description of 60 Imbalanced Datasets.

| Data# | Size | Feat. | IR | Data Name | Area |
|---|---|---|---|---|---|
| *More Imbalanced (20 data)* | | | | | |
| 1 | 403 | 35 | 12.00 | mw1 | Software Engineering |
| 2 | 661 | 37 | 11.71 | PizzaCutter1 | Software Engineering |
| 3 | 365 | 5 | 10.77 | analcatdata_draft | Sports |
| 4 | 705 | 37 | 10.56 | PieChart1 | Software Engineering |
| 5 | 990 | 13 | 10.00 | vowel | Speech/Audio |
| 6 | 504 | 19 | 9.72 | meta | Meta-Learning (Stats) |
| 7 | 458 | 38 | 9.65 | kc3 | Software Engineering |
| 8 | 1320 | 17 | 9.65 | analcatdata_halloffame | Sports |
| 9 | 2000 | 6 | 9.00 | mfeat-morphological | Image Processing |
| 10 | 1043 | 37 | 7.21 | PizzaCutter3 | Software Engineering |
| 11 | 450 | 3 | 7.18 | analcatdata_apnea3 | Medical |
| 12 | 475 | 3 | 6.79 | analcatdata_apnea1 | Medical |
| 13 | 327 | 37 | 6.79 | CastMetal1 | Industrial |
| 14 | 475 | 3 | 6.42 | analcatdata_apnea2 | Medical |
| 15 | 2310 | 17 | 6.00 | segment | Image Processing |
| 16 | 559 | 4 | 5.99 | arsenic-female-bladder | Medical |
| 17 | 470 | 13 | 5.71 | thoracic-surgery | Medical |
| 18 | 381 | 38 | 5.57 | water-treatment | Environmental |
| 19 | 500 | 22 | 5.25 | collins | Medical/Ecology |
| 20 | 562 | 21 | 5.11 | soybean | Agricultural |
| *Mid Imbalanced (20 data)* | | | | | |
| 21 | 1066 | 7 | 4.86 | solar-flare | Physics/Space |
| 22 | 797 | 4 | 4.14 | analcatdata_dmft | Medical/Dental |
| 23 | 522 | 20 | 3.88 | kc2 | Software Engineering |
| 24 | 1324 | 10 | 3.53 | mofn-3-7-10 | Synthetic/Logical |
| 25 | 1156 | 5 | 3.52 | socmob | Social Science |
| 26 | 400 | 5 | 3.44 | analcatdata_germangss | Financial |
| 27 | 812 | 6 | 3.34 | unknown | Industrial |
| 28 | 363 | 8 | 3.27 | braziltourism | Geographic/Business |
| 29 | 748 | 4 | 3.20 | blood-transfusion-service-center | Medical/Logistics |
| 30 | 336 | 14 | 3.10 | primary-tumor | Medical |
| 31 | 846 | 18 | 2.88 | vehicle | Image Processing/Automotive |
| 32 | 1000 | 20 | 2.86 | autoUniv-au1-1000 | Synthetic/Automotive |
| 33 | 306 | 3 | 2.78 | haberman | Medical |
| 34 | 583 | 10 | 2.49 | ilpd | Medical |
| 35 | 1728 | 6 | 2.34 | car | Social Science/Automotive |
| 36 | 1000 | 19 | 2.33 | credit-g | Financial |
| 37 | 358 | 31 | 2.23 | dermatology | Medical |
| 38 | 328 | 32 | 2.22 | analcatdata_marketing | Business/Marketing |
| 39 | 641 | 19 | 2.16 | eucalyptus | Ecology/Forestry |
| 40 | 593 | 77 | 2.14 | emotions | Audio/Psychology |
| *Less Imbalanced (20 data)* | | | | | |
| 41 | 310 | 6 | 2.10 | vertebra-column | Medical |
| 42 | 2201 | 2 | 2.10 | Titanic | History/Social Science |
| 43 | 1074 | 16 | 2.09 | colleges_aaup | Education/Socioeconomic |
| 44 | 973 | 9 | 2.02 | xd6 | Synthetic/Rule-Based |
| 45 | 320 | 6 | 2.00 | pc1_req | Software Engineering |
| 46 | 1055 | 32 | 1.96 | qsar-biodeg | Chemistry/Toxicology |
| 47 | 462 | 9 | 1.89 | sa-heart | Medical |
| 48 | 958 | 9 | 1.87 | tic-tac-toe | Game/AI |
| 49 | 768 | 8 | 1.86 | diabetes | Medical |
| 50 | 683 | 9 | 1.79 | breast-w | Medical |
| 51 | 351 | 33 | 1.78 | ionosphere | Physics/Space |
| 52 | 250 | 12 | 1.77 | Horse Colic | Medical/Veterinary |
| 53 | 959 | 40 | 1.77 | tokyo1 | Image Processing |
| 54 | 609 | 7 | 1.73 | kdd_el_nino-smal | Earth Science/Climate |
| 55 | 569 | 30 | 1.68 | wdbc | Medical |
| 56 | 392 | 8 | 1.67 | cars | Automotive |
| 57 | 349 | 31 | 1.64 | cylinder-bands | Industrial |
| 58 | 250 | 9 | 1.60 | Horse Colic | Medical/Veterinary |
| 59 | 500 | 25 | 1.55 | unknown | Unknown/General |
| 60 | 4601 | 7 | 1.54 | spambase | Information Technology |

IR: Imbalance Ratio (major:minor = X:1)

## A.6 VISUALIZATION - EXTREME IMBALANCE

Figure 8 shows the distribution of extremely imbalanced datasets.

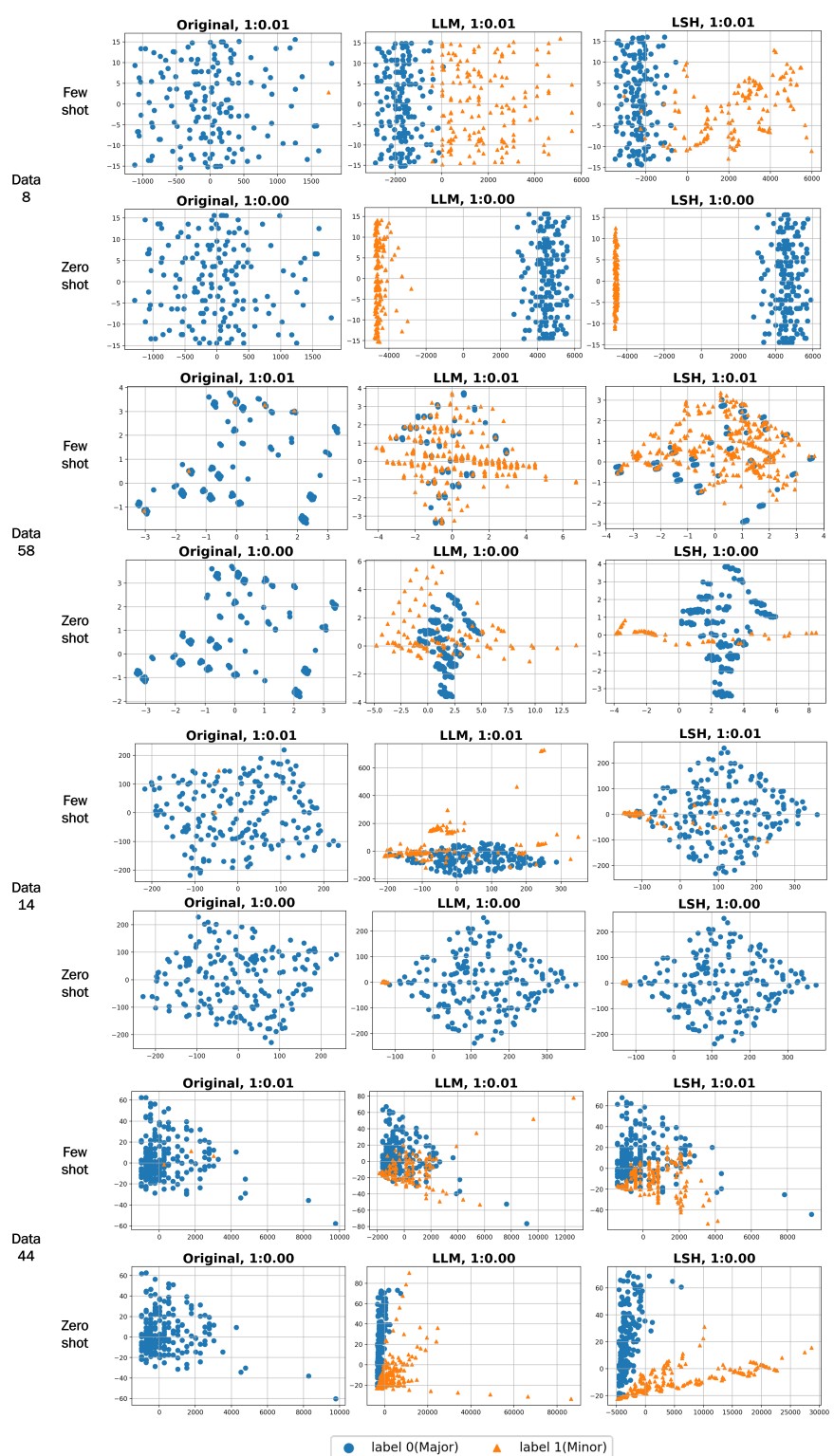

Figure 8: Visualization of the four datasets under few-shot/zero-shot scenarios.

