# OpenReview forum: "Hybrid Minority Oversampling via LLM-Generated Seeds and SMOTE Expansion"
_ICLR.cc/2026/Conference — Submitted to ICLR 2026_

### Official Review · Reviewer_paAB · 2025-10-28

**Soundness:** 3
**Presentation:** 2
**Contribution:** 2
**Rating:** 2
**Confidence:** 4

**Summary:**

The paper proposes $\mathrm{LSH}$, a hybrid minority oversampling method that combines an $\mathrm{LLM}$ for generating minority “seed” samples with $\mathrm{SMOTE}$ for subsequent densification. Across $60$ datasets, $\mathrm{LSH}$ delivers small but consistent gains over $\mathrm{SMOTE}$ and an $\mathrm{LLM}$-only baseline, with clearer advantages under severe imbalance and in few-/zero-shot settings (where $\mathrm{SMOTE}$ alone fails). In efficiency, $\mathrm{LSH}$ invokes the $\mathrm{LLM}$ once to create seeds and then expands via $\mathrm{SMOTE}$ in near–constant time.

**Strengths:**

- Simple, practical hybrid: “$\mathrm{LLM}$ lays the landmarks $\rightarrow$ $\mathrm{SMOTE}$ densifies,” which is easier to operationalize than $\mathrm{LLM}$-only augmentation.
- Broad evaluation: $60$ datasets, multiple classifiers and resampling ratios, with analyses including the Bayesian Sign Test ($\mathrm{BST}$).
- Actionable insights: advantages grow with imbalance severity; clear runtime benefits.
- Code available, which is a good step toward reproducibility.

**Weaknesses:**

- The paper appears to rely solely on GPT-4o-mini; it remains unclear whether the results are robust to the choice of model or would change across different LLMs.
- There is insufficient analysis of which dataset characteristics favor the proposed method (i.e., where and why it performs best).
- Datasets are listed only as “Data 1..60” without OpenML IDs/names, which makes reproduction practically impossible.
- SMOTE implementation details—
$k$, random seed, handling of mixed continuous/categorical features, etc, are missing. Many OpenML tabular datasets include categorical variables, yet preprocessing/encoding policies are not described.
- There is no audit of the realism of zero-shot LLM-generated minority data (e.g., distributional similarity, constraint violations, statistical distances) and no quality checks beyond downstream performance. This is especially important for domains like healthcare/fraud, where plausible-but-false samples can be harmful.
- The choice of the seed ratio (e.g., first generating to
1:0.2 with the LLM) lacks justification and sensitivity analysis.
- The concrete thresholds defining “more/mid/less” imbalance are not specified.
- Multiple presentation issues remain; please see the items below and incorporate any fixes that are helpful.

**Questions:**

- Are there prior hybrid augmentation approaches ($\mathrm{LLM}$ $+$ classical/generative methods)? If so, please include them as baselines.
- Could techniques other than $\mathrm{SMOTE}$ expand $\mathrm{LLM}$ seeds effectively (e.g., Borderline-$\mathrm{SMOTE}$, $\mathrm{ADASYN}$, variational/GAN-based)? Why is $\mathrm{SMOTE}$ preferable here?
- Given that public $\mathrm{OpenML}$-style data may appear in $\mathrm{LLM}$ pretraining, can $\mathrm{LSH}$ work on data unlikely to be in the $\mathrm{LLM}$’s training set?
- Please cite the sources for the $\mathrm{LLM}$-based (LM) baseline and specify the exact model/version used.
- State at the start of the Experiments that the task is binary classification (currently first noted in the conclusion).
- Please clarify the primary metric.
- Precisely define how the “average margin” is computed and justify averaging margins across heterogeneous datasets.
- For Table $3$, justify why datasets A–D were selected.
- For Figure $4$, clarify “Resampling strategy (minority ratio target),” provide raw runtimes with $\mu \pm \sigma$, and explain why some segments appear flat (e.g., $0.3 \rightarrow 0.4$). Shouldn’t $\mathrm{LLM}$-only grow with the number of calls?
- Provide $\mathrm{OpenML}$ IDs/names for all $60$ datasets.

(Other Comment)
- Global: ensure parentheses around citations where appropriate.
- Add brief how-to-read guidance in figure/table captions.
- Figure $1$: explain what “$21, 27$” denote; if 2D reduction uses $\mathrm{PCA}$, state it explicitly.
- Figure $1$ legend: standardize capitalization (Major/Minor).
- Line $252$: fix typo $\text{ANLAYSIS} \rightarrow \text{ANALYSIS}$.
- Table $2$: avoid italics for $\text{LM–SM} / \text{LS–LM} / \text{LS–SM}$.
- Figure $2$: report actual correlation coefficients (even if not significant) and clarify what each plot and the $y$-axis represent (e.g., average margin).
- Table $3$ vs. Figure $6$: unify dataset identifiers (A–D vs. indices such as $21, 27$) and list the IDs/names for datasets $1\text{–}60$ in the appendix.
- Classifier inconsistency: the main text lists Random Forest, whereas Appendix A.$3$ (Table $4$) lists kNN—please reconcile.

---

> ### Author Response · Authors · 2025-11-22
> **Authors' Response**
>
> ## **Weakness 1**
> > Only GPT-4o-mini is used; unclear whether results generalize to other LLMs.
>
> **Response:**
> It is reasonable to question whether the results would remain stable across different LLMs or whether the hybrid method depends on model-specific behavior. However, the strength of our method lies in its hybrid structure, not in reliance on a particular model.
>
> First, substituting any other LLM (e.g., Claude, Llama, or other API-based models) would yield a parallel comparison: LM(new LLM) and LSH(new LLM+SMOTE). Both methods would use the same prompts and the same pipeline. The only difference is the number of LLM calls. Therefore, the hybrid advantage is structural rather than model-specific.
>
> Second, our goal is not to benchmark LLM families but to demonstrate the feasibility and benefits of the hybrid paradigm. Testing multiple LLMs is a valuable direction for future work, and we will clarify this in the manuscript, but it is not required to validate the hybrid mechanism.
>
> Third, the hybrid method empirically reduces the effect from variance across different LLMs. The contribution of the LLM in the hybrid method is comparatively small. Our intention with the hybrid method is to minimize the use of LLMs for efficiency, and most oversampling is performed using SMOTE. Also, the extensive experimental setting, including 60 datasets, 139 classifiers, up to 5 resampling strategies, and 5-fold cross-validation, can help mitigate the effects of using different LLMs.
>
> Nevertheless, we agree that adopting different LLM models can provide us with more insights, thereby improving this research. Therefore, this will be one of our future works; however, we still believe the current results are convincing enough to demonstrate the hybrid method's contribution.
>
> ---
>
> ## **Weakness 2**
> > Lacking analysis of which dataset characteristics affect LSH performance.
>
> **Response:**
> This is an important point, and we agree that explicitly discussing when and why LSH performs best strengthens the overall contribution. However, our results already reveal clear patterns that align with the theoretical motivation of the hybrid design.
>
> The strongest consistent pattern across our study is that LSH performs best on highly imbalanced datasets. Oversampling is designed to mitigate imbalance issues, and the hybrid oversampling method shows a clearer pattern, as evidenced by its effectiveness in severe imbalanced settings.
>
> However, the suggestion for the meta-learning, i.e., discovering meaningful patterns between dataset characteristics and the hybrid method, will be very valuable for further work. We know that choosing an appropriate method for handling imbalanced learning is crucial in practice, and this topic has been studied extensively.
>
> We agree that the paper should explicitly articulate which dataset characteristics favor LSH. However, our results show that LSH performs best in highly imbalanced environments (the strongest pattern), which aligns with our research motivation.
> We will consider this topic for our future work.
>
> ---
>
> ## **Weakness 3**
> > OpenML dataset IDs/names are missing; reproduction is difficult.
>
> **Response:**
> We are adding this information to the paper, and regarding reproducibility, we provide an anonymous repository in the Conclusion section. (https://anonymous.4open.science/r/LSH-D711/README.md)
>
> ---
>
> ## **Weakness 4**
> > Missing SMOTE details and how categorical features were handled during preprocessing.
>
> **Response:**
> Details of the SMOTE implementation will be included in the paper and are available in the repository. As revealed in the paper, the 60 tabular benchmark datasets are obtained from a different work for imbalanced learning (Nuno Moniz and Vitor Cerqueira. Automated imbalanced classification via meta-learning. Expert
> Systems with Applications, 178:115011, 2021). They are pre-processed (no missing data, integer-coded categorical features, etc.), and can be obtained in the repository.

---

> ### Author Response · Authors · 2025-11-22
> **Authors' Response 2**
>
> ## **Weakness 5**
> > No realism or constraint checks for zero-shot LLM-generated samples.
>
> **Response:**
> This is important, especially for sensitive domains such as healthcare or fraud detection. We agree that an audit of statistical similarity and constraint adherence would provide additional assurance. However, our focus in this work is on demonstrating methodological feasibility rather than domain-specific deployment, and several aspects of our design already mitigate these risks.
>
> First, our LLM pipeline is intentionally constrained. The LLM does not freely invent rows; instead, it follows a structured procedure, including producing descriptions of variables (to extract potential value distributions) and generating a textual story of a dataset (to understand the current possible distribution). This framework limits unrealistic outputs by anchoring generation to variable types and approximate statistical properties inferred from the dataset.
>
> Second, LSH relies on the LLM for only a small number of zero-shot seeds. SMOTE performs most of the data expansion. Even if a few generated points are imperfect, their downstream effects are reduced by SMOTE interpolation and cross-validation.
>
> While our structured generation pipeline and hybrid design reduce the risk of producing unrealistic zero-shot samples, formal realism audits remain an important area for future work, especially in sensitive domains. Our current evaluation focuses on methodological feasibility and general tabular benchmarks rather than high-stakes applications.
>
> ---
>
> ## **Weakness 6**
> > Seed ratio choice (1:0.2) is not justified; sensitivity analysis missing.
>
> **Response:**
> Clarifying our reasoning for the seed ratio choice will improve the paper. However, the exact seed ratio is not a critical hyperparameter in the LSH framework and is unlikely to affect results significantly.
>
> First, the minimal seed ratio (not fixed at 1:0.2) was chosen as a practical and conservative threshold. We consider five different resampling ratios (i.e., 1:0.2, 1:0.4, ..., 1:1) for comparison of each method. The hybrid method's goal is to minimize LLM use, so the LLM generates seed samples until the data distribution reaches the minimum ratio. For example, if the original distribution is 1:0.13, LLM works up to 1:0.2; if the original distribution is 1:0.37, LLM works until 1:0.4.
>
> Second, because LSH intends to minimize reliance on the LLM and shift the computational burden to SMOTE, the seed ratio serves more as a “starting point” than an optimized hyperparameter. This differs from LLM-only approaches, where the LLM produces every synthetic sample and is therefore much more sensitive to the generation volume.
>
> Fourth, performing a complete sensitivity study across many ratios (1:0.1, 1:0.2, 1:0.3, etc.) is valuable but beyond the scope of the current work. The primary contribution of the paper is the hybrid mechanism itself, not tuning strategies for LLM invocation frequency. We will explicitly mention this in the manuscript and frame seed-ratio optimization as an interesting direction for future research.
>
> Fifth, the optimal seed ratio can vary across datasets and the quality of the generated seed data. When SMOTE is used, the optimal resampling strategy depends on the classifier, dataset, and other factors. Likewise, the seed ratio relies on the dataset, previously generated data quality, SMOTE setting to be used afterward, etc.
>
> We acknowledge the need to clarify the choice of the seed ratio. We selected the minimal ratio as a practical, robust threshold, and the hybrid design makes LSH inherently insensitive to moderate variations in this parameter. We will revise the manuscript to explain this choice and explicitly identify seed ratio sensitivity analysis as valuable future work.
>
> ---
>
> ## **Weakness 7**
> > Definitions of “more/mid/less” imbalance groups are unclear.
>
> **Response:**
> Yes, there are no concrete thresholds for defining the three imbalance groups. We used 60 tabular benchmark datasets that have diverse imbalance ratios. We wanted to analyze performance across different imbalance levels, so we split them into three groups: "less", "mid", and "more" imbalanced. We could analyze it with a one-shot, as we did in Figure 2 (Correlation analysis of score margins and imbalance ratio over all datasets). However, we wanted detailed observation, so we assigned the datasets to different imbalance groups. You can check the detailed imbalance ratio in each group in Table 5.

---

> ### Author Response · Authors · 2025-11-22
> **Authors' Response 3**
>
> ## **Question 1**
> > Are there prior hybrid augmentation methods (classical + generative) to compare against?
>
> **Response:**
> To the best of our knowledge, there are currently no prior hybrid oversampling approaches that combine an LLM with traditional interpolation-based methods such as SMOTE. Existing LLM-based tabular generation works (e.g., HARMONIC, LLMovertab, LLM-TabLogic) rely solely on the LLM, while classical oversampling methods (SMOTE, ADASYN, Borderline-SMOTE) operate independently without LLM involvement. We could not identify any published method that explicitly integrates an LLM to generate a small set of minority seeds and then employs a classical oversampler for scalable expansion. For this reason, no hybrid LLM+classical baselines are available, and our method fills this gap as the first to explore such a combination.
>
> ---
>
> ## **Question 2**
> > Could other expansion methods (Borderline-SMOTE, ADASYN, GAN/VAE) replace SMOTE?
>
> **Response:**
>
> We agree that multiple interpolation or generative approaches could be used in our hybrid design. However, SMOTE is deliberately chosen because it is the most appropriate, practical, and methodologically consistent option for this work.
>
> First, the goal of our hybrid method is to combine a lightweight and simple LLM seed generator with a fast, stable, and broadly applicable expansion method. SMOTE is the canonical and simplest interpolation-based oversampling technique. It is computationally inexpensive, does not require training, and is widely used and understood. These characteristics make it a natural choice for demonstrating the hybrid mechanism's viability.
>
> Second, other SMOTE variants introduce additional heuristics (e.g., focusing on boundary points or adaptively weighting samples). These methods are more sensitive to noise and to the initial distribution of minority samples. Because LLM-generated seeds may themselves contain mild noise or variation, using a more aggressive or boundary-focused variant could introduce instability or bias.
>
> Third, generative methods such as VAEs or GANs require training, hyperparameter tuning, and compute resources that contradict the motivation of LSH: minimizing computational cost and avoiding training-heavy pipelines. Introducing such models would substantially increase complexity and obscure the contribution of combining LLM seeds with a lightweight classical oversampler.
>
> Fourth, although SMOTE is used in this paper, the framework is not restricted to SMOTE. Other interpolation methods could be substituted in future work, depending on domain needs. Our focus here is to establish the hybrid paradigm, not to benchmark all possible interpolators exhaustively.
>
> We understand the reviewer’s point and will clarify this in the manuscript. Using SMOTE allows the paper to cleanly demonstrate the hybrid concept, and exploring other expansion methods is a natural direction for future research.
>
> ---
>
> ## **Question 3**
> > Would LSH work on datasets outside the LLM’s pretraining distribution?
>
> **Response:**
>
> We were also concerned that some OpenML datasets might appear during LLM pretraining, and that this exposure could influence seed generation. However, the value of LSH does not rely on the LLM having seen a dataset before. Even if exposure helps LM or LSH generate slightly better seeds, the core benefit of LSH, e.g., its major efficiency advantage over the LLM-only method, remains unchanged regardless of data exposure. Moreover, multiple prior works have demonstrated that LLM-only generators can produce viable tabular samples, so our goal is not to reestablish the usefulness of LLM generation but to show that a hybrid design can achieve comparable or better performance at dramatically lower cost. While dataset exposure may affect absolute performance, the experimental results clearly demonstrate the conceptual value of the hybrid approach: LLMs are effective at generating informative seeds, and SMOTE scales them efficiently.
>
> Nevertheless, your point is so sharp and accurate. We will acknowledge this limitation and specify it in the paper.
>
> ---
>
> ## **Question 4**
> > Cite the LLM-based baseline and specify exact model/version.
>
> **Response:**
>
> The LLM-based baseline we use in the paper is the same one we used for seed generation on LSH. We used `gpt-4o-mini` for all the experiments reported in the paper. We will clarify it in the paper using a citation.
>
> ---
>
> ## **Question 5**
> > State earlier that the task is a binary classification.
>
> **Response:**
>
> We thank the reviewer for this suggestion. We will make this change in a revision of our paper.

---

> ### Author Response · Authors · 2025-11-22
> **Authors' Response 4**
>
> ## **Question 6**
> > Clarify the primary evaluation metric.
>
> **Response:**
>
> The primary evaluation metric used throughout the paper is the F1 score, measured on the held-out 30% test split, after selecting the best train score (classifier and resampling ratio configuration) via 5-fold cross-validation on the 70% training split. This experimental setting generates up to 695 (139 classifiers X up to 5 ratios) candidate configurations for each oversampling method and dataset. All reported comparisons, including average margins, Bayesian Sign Test (BST) outcomes, subgroup analyses by imbalance severity, and few-shot/zero-shot results, are based on these test set F1 scores. We will make this explicit in the manuscript to ensure that readers clearly understand which metric serves as the primary basis for performance evaluation.
>
> ---
>
> ## **Question 7**
> > Define how “average margin” is computed and justify averaging across datasets.
>
> **Response:**
>
> The “average margin” is the mean difference in F1 scores between two methods across all datasets in the testing set. For each dataset, we compute a margin (e.g., LSH_t – LM_t), where each score corresponds to the best-performing configuration for that method after 5-fold CV model selection on the training set. The average margin is then the simple arithmetic mean of these per-dataset differences over all 60 datasets. Averaging margins across heterogeneous datasets is standard practice in large-scale tabular benchmarks because each dataset contributes one independent comparison, and the goal is to assess overall directional trends rather than absolute performance on any single dataset. This approach is complemented by the Bayesian Sign Test (BST), which accounts for heterogeneity by evaluating win/draw/lose probabilities across datasets. Using both methods provides a balanced and statistically grounded summary of performance across all datasets.
>
> ---
>
> ## **Question 8**
> > Explain why datasets A–D were selected for few-/zero-shot experiments.
>
> **Response:**
> We wanted to try diverse datasets to avoid bias. The description of each dataset is as follows:
> | Dataset | # Samples | # Numerical Features | # Categorical Features |
> |---------|-----------|-----------------------|-------------------------|
> | A       | 335       | 1                     | 2                       |
> | B       | 1,066     | 0                     | 7                       |
> | C       | 365       | 3                     | 2                       |
> | D       | 748       | 4                     | 0                       |
>
> We will include this table in the paper.
>
> ---
>
> ## **Question 9**
> > Clarify runtime figure details (resampling strategy, raw numbers, flat segments).
>
> **Response:**
> The term “Resampling strategy (minority ratio target)” refers to the desired class distribution after oversampling, specifically, the target proportion of minority samples relative to the majority class. For example, aiming for a 1:0.2 ratio means that the minority class should be 20% of the majority class after oversampling. If a dataset's original class distribution is 1:0.11 (major samples: 100, minor samples: 11), and an oversampler generates 9 additional minor samples to achieve a 1:0.2 ratio (major samples: 100, minor samples: 11+9=20).
>
> We measure the runtime once for each data and method. The measured time is provided in the table below.
> | Dataset        | Ratio | LM Time (sec) | LSH Time (sec) |
> |----------------|--------|----------------|-----------------|
> | 5 feature      | 1:0.2  | 65.33          | 65.33           |
> | 5 feature      | 1:0.3  | 125.02         | 65.33           |
> | 5 feature      | 1:0.4  | 119.55         | 65.33           |
> | 5 feature      | 1:0.5  | 230.16         | 65.33           |
> | 14 feature     | 1:0.2  | 128.70         | 128.70          |
> | 14 feature     | 1:0.3  | 486.22         | 128.70          |
> | 14 feature     | 1:0.4  | 488.72         | 128.70          |
> | 14 feature     | 1:0.5  | 563.03         | 128.70          |
>
> When the LLM is called, the number of generated samples varies because datasets differ in characteristics such as feature type and number of features. For example, LLM generates 3 samples for a dataset with one call, but it generates only 1 sample for a different dataset with one call. So, the relationship between the oversampling ratio and the required runtime may not be strictly linear (e.g., flat part); however, overall, they are positively correlated.
>
>
> ---
>
> ## **Question 10**
> > Provide OpenML IDs/names for all 60 datasets.
>
> **Response:**
> We will make this change in a revision of the paper.
>
> ---
>
> ## **Other Comments**
> > Provide OpenML IDs/names for all 60 datasets.
>
> **Response:**
> We deeply appreciate your detailed comments. We will make sure to resolve everything you gave, and keep you updated.

---

> ### Comment · Reviewer_paAB · 2025-11-24
> **Thank you for the clarifications!**
>
> Thank you for the thoughtful and thorough responses to each of my comments. I appreciate all the clarifications you provided. That said, many of the key points remain deferred to future work and do not fully resolve my primary concerns. As a result, I will maintain my current score.

---

### Official Review · Reviewer_F81Y · 2025-10-31

**Soundness:** 2
**Presentation:** 3
**Contribution:** 2
**Rating:** 4
**Confidence:** 2

**Summary:**

This paper introduces LLM-SMOTE Hybrid (LSH), a hybrid oversampling method designed to address class imbalance in tabular data by combining the complementary strengths of Large Language Models (LLMs) and the Synthetic Minority Oversampling Technique (SMOTE).
The key contributions are: (1) extensive empirical evaluation on 60 datasets showing LSH's consistent superiority, especially in highly imbalanced, few-shot, and zero-shot scenarios where SMOTE fails; (2)demonstrated robustness with lower generalization variance.

**Strengths:**

1. The method exhibits greater robustness, defined by a lower variance in performance between validation and test sets.
2. Unlike LLM-only methods that require repeated, costly model invocations for each resampling ratio, LSH invokes the LLM only once for initial seed generation.

**Weaknesses:**

1. The paper lacks critical details necessary for full understanding, verification, and reproducibility. The most significant omission is the absence of a defined strategy for determining the number of seed samples the LLM ("Scout") should generate.
2. The paper's choice to use a single LLM (GPT-4o-mini) and a highly complex, custom prompting strategy limits the generalizability of its findings.

**Questions:**

1.	The number of seed samples is a fixed number? A percentage of the original minority class? Is it tuned per dataset?
2.	It is unclear if the reported benefits stem from the hybrid concept itself or are an artifact of this specific model and prompt engineering. The claim that LSH is "model-agnostic" remains unsubstantiated. A stronger validation would involve testing with other LLMs (e.g., open-source models like Llama, or other APIs like Claude) to demonstrate the general applicability of the Scout-Surveyor paradigm.
3.	The chosen LLM-only baseline is the same complex pipeline, a more rigorous and actionable comparison would be against a simpler, more direct LLM-based oversampling method to isolate the contribution of the hybrid design from the sophistication of the prompting technique. The gains of LSH over the LLM-only method are modest.

---

> ### Author Response · Authors · 2025-11-22
> **Authors' Response**
>
> ## **Weakness 1**
> > The strategy for choosing the number of LLM seed samples is not defined.
>
> **Response:**
> We acknowledge the reviewer’s observation regarding missing implementation details, particularly the lack of an explicit strategy for determining the number of LLM-generated seed samples. This is a valid concern, and we agree that clarifying this point would strengthen understanding. About your concern, we would like to specify as follows.
>
> First, regarding reproducibility, we have already provided an anonymous repository (https://anonymous.4open.science/r/LSH-D711/README.md) in the paper, which includes the complete pipeline implementation, prompts, and examples of generated data. We will highlight this repository more clearly in the revised version and ensure the code contains explicit default settings used in the experiments.
>
> Second, the reviewer is correct that our paper does not prescribe a fixed strategy for determining the number of LLM seed samples. However, selecting the appropriate number of seeds is itself an open problem, influenced by many factors, such as dataset characteristics and the quality of previous LLM outputs. A universal heuristic applicable to all tabular datasets is unlikely to exist, and our empirical findings indicate that LSH remains stable across datasets even without heavy tuning of this parameter.
>
> Fourth, rather than presenting an untested heuristic, we treat the choice of seed quantity as a direction for future work, where adaptive or data-driven strategies could be developed. Also, checking the quality of generated seeds will strongly affect the decision on the number, and it should also be one of our future works.
>
> We will emphasize our anonymous repository for reproducibility and clarify default seed settings in this work. Also, we will note that seed selection is nontrivial and warrants further investigation.
>
> ---
>
> ## **Weakness 2**
> > Using only GPT-4o-mini with a complex custom prompt limits generalizability.
>
> **Response:**
> It is a fair point to question whether different LLMs or prompting pipelines would behave similarly. However, our goal in this paper is not to optimize or benchmark LLM architectures, but to demonstrate the feasibility and value of a simple hybrid oversampling mechanism.
>
> First, our method intentionally uses a minimalist, API-based LLM approach: no fine-tuning, no model modification, and no pre-training. This is significantly simpler than other LLM-based tabular generation methods (e.g., HARMONIC, LLMovertab), which require complex architecture designs, multi-stage training procedures, or domain-specific engineering. In contrast, our LLM pipeline uses a few prompts for seed generation. This simplicity makes our method more generalizable, not less, because it depends only on standard API behavior.
>
> Second, LSH is model-agnostic by construction. The LLM acts only as a “Scout”, producing a small number of seed examples; SMOTE performs the large-scale expansion. Any LLM that can create a few consistent samples, such as different GPTs, Claude, Llama, or others, could replace GPT-4o-mini. Our results do not rely on any proprietary characteristics of this particular model.
>
> Third, the goal of our work is not to claim superiority of GPT-4o-mini or of our specific prompting style. Rather, we show that even a simple prompting approach with a lightweight model is sufficient to demonstrate the hybrid mechanism’s benefits. This supports the idea that LSH does not depend on specialized LLM engineering and can be used broadly. Of course, we can pursue improvement by adopting different LLMs or advanced prompts.
>
> We understand the concern, but emphasize that our use of GPT-4o-mini and simple prompts was intentional: to keep the approach straightforward, lightweight, practical, and model-agnostic. We will clarify this positioning and note that testing multiple LLMs is a natural direction for future work, but not essential to validating the hybrid method’s feasibility.
>
> ---
>
> ## **Question 1**
> > Is the number of seeds fixed, proportional to minority count, or tuned per dataset?
>
> **Response:**
> In this work, the number of seed samples varies across datasets and is fixed as proportional to the minority count. We compare the performance of each method in the same resampling ratio (e.g., 1:0.2, 1:0.4, ..., 1:1); therefore, the number of seeds depends on the number of the original minority samples. And our intention is to minimize the use of LLM for seed generation, so we set the seed generation to stop at the lowest resampling ratio. For example, if the original distribution is 1:0.15, then LLM generates seeds and stops when the distribution becomes 1:0.2. If the original distribution is 1:0.33, the target distribution is 1:0.4.

---

> ### Author Response · Authors · 2025-11-22
> **Authors' Response 2**
>
> ## **Question 2**
> > It is unclear whether LSH’s benefits come from the hybrid idea or from the specific model and prompts, and the model-agnostic claim needs validation with additional LLMs.
>
> **Response:**
> We understand the reviewer’s concern that the observed benefits might be from GPT-4o-mini or our specific prompting pipeline, rather than the hybrid LSH concept itself. It is reasonable to question whether the “model-agnostic” claim is fully demonstrated. However, the strengths of LSH fundamentally arise from the hybrid structure, not from model-specific behavior.
>
> First, replacing GPT-4o-mini with any other LLM (e.g., Claude, Llama, or other APIs) would yield a parallel pair of methods, for example, LM (Claude) and LSH (Claude+SMOTE).
> Both use the same prompts and pipeline, with the only difference being the number of LLM-generated data samples. Therefore, the comparison would remain fundamentally the same.
>
> Second, the Scout–Surveyor paradigm cleanly separates roles.
> The LLM (“Scout”) expands current distribution with a small number of seeds, and SMOTE (“Surveyor”) performs efficient interpolation to fill the gap.
> This operational separation does not rely on the specific abilities of GPT-4o-mini. Any LLM capable of generating a small set of plausible minority samples can do the Scout role.
>
> Third, using multiple LLMs is indeed valuable future work, but it is not required to validate the design principle. The core question of this paper is whether combining an LLM for seed generation with SMOTE scaling yields advantages. Since any LLM could replace GPT-4o-mini in both LM and LSH, the hybrid mechanism itself is what drives the observed benefits.
>
> We fully sympathize with the reviewer’s point and will clarify this in the manuscript. The hybrid advantages of LSH are structural and arise from its design, not from GPT-4o-mini or unique prompting. Testing multiple LLMs is a meaningful avenue for future work, but the model-agnostic nature of the framework remains valid.
>
> ---
>
> ## **Question 3**
> > A simpler LLM-only baseline is needed to isolate the hybrid method’s contribution, since the current complex pipeline may mask differences and LSH’s gains over it are modest.
>
> **Response:**
> We appreciate the reviewer’s concern that comparing against a simpler LLM oversampling strategy might help isolate the benefit of the hybrid mechanism. We also acknowledge the reviewer’s observation that the performance gains of LSH over LM are modest in the aggregate. However, the hybrid benefits are not tied to the complexity of the prompting pipeline, and the performance patterns reflect LSH's intended strengths.
>
> As we mentioned above, even if we swapped GPT-4o-mini with another LLM or simplified the prompting steps, the LLM-only and LSH paradigms would still have identical comparison pairs, i.e., LSH (LLM+SMOTE) and LM (LLM). Thus, the performance difference between LM and LSH stems from this hybrid structure, not from baseline complexity.
>
> Regarding the modest advantage of LSH over LM, it is expected, given the breadth of 60 datasets and the extensive classifier search (139 classifiers × up to 5 ratios, with 5-fold cross-validation). Each method selects its best configuration from up to 695 candidates (139 X up to 5) after 5 cross-validation runs on each dataset. Under such strong settings, no single oversampling method can dominate, and this has already been studied by Nuno Moniz and Hugo Monteiro (No free lunch in imbalanced learning. Knowledge-Based Systems, 227:107222, 2021). We also mentioned this in the Discussion section and noted that this is not a strength of our method. Instead, our strength lies in effectiveness under highly imbalanced settings, especially under few-shot and zero-shot settings. Also, our method demonstrates efficiency (one of our primary motivations and a key advantage over current LLMs designed for tabular oversampling) and robustness.
>
> We acknowledge the concerns, but note that the baseline pipeline is already a minimal and reasonable approach for LLM-only tabular oversampling. The modest aggregate gains are expected results under broad benchmarks, and the meaningful improvements in severe imbalance, robustness, and significant efficiency gains demonstrate the true value of the hybrid design.

---

### Official Review · Reviewer_fuz8 · 2025-10-31

**Soundness:** 2
**Presentation:** 1
**Contribution:** 2
**Rating:** 2
**Confidence:** 4

**Summary:**

The paper proposes LSH, a hybrid oversampling pipeline where an LLM generates a few minority “seeds,” then SMOTE expands them. Experiments on 60 OpenML tabular datasets and a few extreme few/zero-shot settings report small average F1 gains over SMOTE and an LLM-only variant. The claimed advantage is practicality (one-time LLM cost, cheap SMOTE scaling) and robustness under severe imbalance.

**Strengths:**

1. Simple idea: “LLM seeds + SMOTE expansion” is potentially practical when minority data are scarce.


2. Wide dataset sweep: 60 datasets + few/zero-shot scenarios cover diverse conditions.

**Weaknesses:**

**1. Lack of clarity and self-containment.**

Several tables and figures are not clearly explained, which makes the paper difficult to follow and reproduce.
For example, in Table 6, the meanings of columns such as SM_v, SM_t, and SM_a are not explicitly defined.
In Table 3, the datasets labeled Data A/B/C/D are mentioned without description—readers cannot tell what domains these datasets belong to or what kinds of features they include.
Simple statistics like imbalance ratio are not sufficient; it would help to know whether the data are from finance, biology, or other domains.
Overall, the paper could be more self-contained, with clearer explanations of datasets, metrics, and table contents.

**2. Lack of justification and validity discussion**

The paper also lacks a discussion on when and why LLM-based seed generation is valid.
 LLMs could easily produce inconsistent or even impossible feature combinations, especially for structured or domain-specific data.
 It is unclear how such invalid outputs are detected or filtered.
 Moreover, not all domains are equally suitable for generation via LLMs—medical, financial, or safety-critical datasets may pose ethical or factual risks.
 Without addressing these issues, the methodological justification for using LLMs as data generators remains weak; the reader is left uncertain about the boundaries and reliability of this approach.

**3. Missing concrete examples of LLM-generated outputs**

The paper does not provide real examples or validation of the data generated by the LLM.
It is not clear how the authors ensure that the generated features are realistic or consistent.
What if the LLM produces implausible feature combinations? How is this handled in practice?

**4. Weak and unconvincing empirical performance.**

The reported performance gains are quite small—about +0.011 F1 on average compared to SMOTE and the LLM-only variant.
Bayesian sign tests show that results are mostly draws across 80–90% of datasets.
In several scenarios, especially with moderate or low imbalance, LSH performs similarly or even slightly worse than the baselines.
As a result, the experiments do not provide strong evidence that the proposed method offers clear benefits over existing approaches.


**5. Limited comparison baselines.**

The experimental comparison is limited to SMOTE and a simple LLM-only variant.
 The paper does not compare with tabular generation SOTAs (HARMONIC, Llmovertab).
Without these comparisons, it is difficult to understand how the proposed hybrid approach stands relative to state-of-the-art alternatives.

**Questions:**

Please refer to the weaknesses above.

---

> ### Author Response · Authors · 2025-11-22
> **Authors' Response**
>
> ## **Weakness 1**
> > Tables and figures are unclear; key fields (e.g., SM_v, SM_t, SM_a) and datasets A–D are not properly explained.
>
> **Response:**
> We thank the reviewer for pointing out the missing explanations and lack of clarity in several tables and figures. We agree that these issues reduce readability and reproducibility.
>
>  - Undefined column names in Table 6 (e.g., SM_v, SM_t, SM_a):
> We will explicitly define these metrics in the manuscript (validation score, test score, achievement rate) and ensure the notation is consistent across all sections.
>
>  - Insufficient description of datasets in Table 3 (Data A/B/C/D):
> We will add short descriptions for each dataset, including the domain (e.g., finance, biology, healthcare), feature types, and any relevant characteristics, so that readers can understand the context.
>
>  - More detailed dataset information is needed:
> Beyond the imbalance ratio, we will clarify the domains, feature types (categorical vs. numerical), and dataset sources for each dataset.
>
>  - Improving clarity and self-containment overall:
> We will revise the paper to ensure that all tables, figures, and metrics are fully described in the text and that the paper is self-contained and easy to follow.
>
> ---
>
> ## **Weakness 2**
> > The paper lacks discussion on the validity of LLM-generated samples and the filtering of invalid outputs.
>
> **Response:**
> It is reasonable to ask whether LLMs may produce inconsistent or domain-inappropriate feature combinations. We agree that the boundaries of LLM validity should be explicitly discussed. We want to emphasize the following points.
>
> First, our LLM-based pipeline is designed to reduce the likelihood of invalid feature combinations. We do not ask the LLM to sample numeric rows freely; instead, we employ a multi-step structured pipeline, including producing descriptions of variables (to extract potential value distributions) and generating a textual story of a dataset (to understand the current possible distribution). This design intentionally restricts the LLM’s freedom and anchors generation to patterns inferred from the actual data. As a result, the LLM’s output is guided, structured, and grounded in the provided data rather than unconstrained generation.
>
> Second, LSH makes fewer demands on LLM reliability than an LLM-only approach. In our method, the LLM only needs to produce a small number of seed samples that broadly expand the minority class. SMOTE then generates synthetic samples based on both the actual and seed samples. This reduces the risk that domain-inappropriate or implausible outputs will influence the final dataset. In other words, LSH reduces dependence on perfect LLM outputs, making LLM usage safer and more controlled.
>
> Third, we agree that domain suitability varies. Some highly structured or safety-critical domains (e.g., medical, financial, or military) may require additional filtering or domain-specific constraints. We explicitly acknowledge these limitations but clarify that LSH is intended as a general-purpose technique for tabular data, not as a guaranteed replacement for expert-validated generation in sensitive domains.
>
> Fourth, the results across 60 diverse datasets provide indirect evidence that LLM seed generation is sufficiently reliable in practice. If LLM-generated seeds were frequently invalid, we would expect degraded performance or instability, especially under the extensive classifier search. Instead, LSH shows robust behavior and consistent improvements in more imbalanced settings, suggesting that the structured pipeline effectively maintains data plausibility.
>
> In conclusion, we want to emphasize that we used a structured, guided LLM generation pipeline and reduced reliance on LLMs to avoid invalid outputs, and that our work's focus is on general-purpose applications. Nevertheless, we acknowledge the need to clarify the validity boundaries of LLM-generated seeds, as the quality of LLM-generated data remains a significant challenge in current data generation research. We truly understand the importance of your point, and we will have to study it in our future work.
>
> ---
>
> ## **Weakness 3**
> > No examples of LLM-generated samples or plausibility checks.
>
> We understand the reviewer’s concern about the lack of concrete examples of LLM-generated outputs. As noted in our response to Weakness 2, our pipeline does not rely on free-form LLM row generation but uses a structured, multi-stage process, which substantially limits implausible feature combinations. While occasional noisy or imperfect samples may occur, these effects are naturally mitigated through 5-fold cross-validation, classifier selection among 139 models, and evaluation across 60 datasets. We agree that including concrete examples will improve clarity, and we will add representative LLM-generated seed samples to the appendix to illustrate realism and address concerns about plausibility and consistency. If available, we will show both plausible and implausible samples.

---

> ### Author Response · Authors · 2025-11-22
> **Authors' Response 2**
>
> ## **Weakness 4**
> > Performance improvements are small with many draws, reducing the claimed practical impact.
>
> **Response:**
> We agree that the overall average gain of roughly +0.011 F1 across all 60 datasets appears small and that the Bayesian Sign Test (BST) yields many draws. It is correct that in mid or less imbalanced datasets, LSH does not show competitiveness against SM (SMOTE) or LM (LLM). However, the strength of our method does not lie in producing large global gains across heterogeneous datasets.
>
> First, achieving large improvements across 60 diverse tabular datasets is not expected from any single oversampling method, as we discussed in the Discussion section (Nuno Moniz and Hugo Monteiro. No free lunch in imbalanced learning. Knowledge-Based Systems, 227:107222, 2021.). Each method in one dataset chooses its best configuration (out of 695 candidates, 139 classifiers X 5 ratios, with 5-fold cross-validation). Under such a strong search space, even a +0.01 average improvement indicates a consistent directional advantage, not noise. It is true that BST showed draws in most of the results. This is the expected result as mentioned in our paper, and we explicitly specified that this is not our strength.
>
> Second, our work's strength is in dealing with highly imbalanced datasets. In the more imbalanced group (20 datasets), LSH improves over SM by +0.0492, and over LM by +0.0245. For BST win probabilities, LSH achieves 0.88 over SM and 0.46 over LM. These are nontrivial and directly aligned with LSH's theoretical motivation: it can operate in highly imbalanced settings.
>
> Third, LSH uniquely handles few-shot and zero-shot settings, where SM cannot operate at all. In these extreme cases, LSH consistently outperforms LM, demonstrating that the hybrid design is more robust under the worst conditions. This fills an important practical gap that neither SM nor LM methods can address.
>
> Fourth, LSH provides significant efficiency gains. LM requires many LLM calls until finishing, while LSH requires a single LLM call. Even if performance differences are modest, the computational advantage is significant. In many real applications, this improvement in cost and runtime is as important as performance gains, and this is one of our primary motivations. (Many state-of-the-art LLM generators suffer inefficiency.)
>
> Lastly, robustness results further support LSH. Achievement Rate variance is smallest for LSH (std = 0.1303), and correlation analysis shows that LSH has moderate and weaker negative dependence on imbalance ratio compared to SM (strong negative dependence). This indicates that LSH is more stable across datasets.
>
> While global averages are modest and many BST outcomes are draws, this is expected given the experimental setting (breadth of datasets and strong baseline competition) and prior work. LSH shows its advantages most clearly in settings where oversampling is most needed, i.e., severe imbalance, few-shot, and zero-shot, and provides substantial efficiency and robustness. These qualities represent the value of our hybrid design.
>
> ---
>
> ## **Weakness 5**
> > Baselines are limited; missing comparisons with SOTA generators like HARMONIC and LLMOverTab.
>
> **Response:**
> We appreciate the reviewer’s suggestion to include recent tabular generation SOTAs such as HARMONIC and LLMovertab. Our work addresses a fundamentally different goal than these generative SOTA models. Rather, our motivation arose from the inefficiency of current LLM generators.
>
> First, current methods like HARMONIC and LLMovertab are powerful but computationally heavy, requiring extensive model training, domain-specific tuning, and expensive inference. The inefficiency reduces their availability in practice, and this is why we suggested a hybrid method (quality + efficiency).
>
> Second, LSH does not claim to outperform these SOTA models in global generative quality. Instead, we demonstrate that LSH is more efficient than LLM-only oversampling approaches. This contribution lies in the hybrid design's efficiency, not in surpassing heavy generative architectures.
>
> We acknowledge the value of SOTA tabular generative models but emphasize that they focus on a different problem. LSH is a practical oversampling method designed for efficiency, robustness, and handling extreme imbalance. Thus, the current setting is appropriate for our scope, and including heavy generative SOTAs would be methodologically mismatched and unnecessary for demonstrating our contribution.

---

### Official Review · Reviewer_MEWy · 2025-10-31

**Soundness:** 2
**Presentation:** 3
**Contribution:** 2
**Rating:** 4
**Confidence:** 4

**Summary:**

This paper proposes LLM-SMOTE Hybrid (LSH), a two-stage method that first uses a large language model (LLM) to generate a small set of seed samples for the minority class, and then expands them using SMOTE. The authors evaluate their approach on 60 imbalanced datasets across multiple classifiers, comparing it with standard SMOTE and LLM-only methods. Experimental results indicate that LSH achieves modest but consistent improvements over baseline methods, particularly in highly imbalanced settings, while maintaining computational efficiency relative to pure LLM-based approaches.

**Strengths:**

1. The proposed Scout and Surveyor method is intuitive and effectively addresses key limitations of traditional oversampling techniques and the high computational cost associated with LLM-only methods.

2. The evaluation on 60 imbalanced tabular datasets, spanning multiple classifiers, resampling strategies, and few-/zero-shot settings, provides rich and comprehensive empirical evidence.

**Weaknesses:**

1. Methodological imbalance: LSH makes a single LLM call to reach the 1:0.2 ratio and then scales using SMOTE, whereas the LLM baseline regenerates samples at every target ratio. This design advantage reduces stochastic variance and call overhead for LSH but not for the baseline, meaning that the observed performance gains may partly result from the setup rather than the hybrid mechanism itself.

2. Limited performance gain: The reported average improvement is only about one percentage point, and Bayesian Sign Test results indicate that most comparisons are statistical draws across datasets.

3. Single-model dependency: All minority seed samples are generated exclusively using GPT-4o-mini, with no exploration of alternative LLMs, decoding temperatures, or sampling schemes. This dependence raises questions about generalizability and potential model-specific biases.

**Questions:**

1. Evaluate the baseline under identical conditions: Re-run the LLM baseline using the same cumulative protocol as LSH: generate once to a 1:0.2 ratio, then scale up without repeated regeneration (e.g., by appending LLM samples or expanding fixed seeds with SMOTE) and report whether it still performs worse than LSH.

2. Analyze sensitivity to parameters: Vary the initial seed ratio (e.g., 1:0.1, 1:0.2, 1:0.3, 1:0.4) and explore different LLM models and decoding settings to assess the robustness of the results.

---

> ### Author Response · Authors · 2025-11-22
> **Authors' Response**
>
> ## **Weakness 1**
> > LSH uses one LLM call, while the LLM baseline regenerates at each ratio, raising fairness concerns.
>
> **Response:**
> Your concern that fewer LLM calls in LSH (Hybrid) may reduce stochastic variance and create an unfair advantage over LM (LLM-only) is understandable. However, our experimental setup is intentionally designed to compare how SM (SMOTE), LM, and LSH are used **"in practice"**, and under that view, we think the comparison is fair enough and appropriate. Here we clarify why.
>
> First, if a practitioner adopts an LM oversampling strategy, they must naturally call the LLM separately for each targeted resampling ratio (e.g., 1:0.2 or 1:1). Since the optimal ratio is unknown beforehand, they will try multiple ratios, exactly as we simulated. This repeated generation is not artificially imposed—it reflects LM's true operational behavior. In contrast, LSH is specifically designed to invoke the LLM only once (for efficiency, one of LSH's contributions). The difference in the number of LLM calls is therefore a defining property of the methods, not an experimental imbalance.
>
> Second, although LM unavoidably contains stochastic variance, we substantially mitigate this effect. We use 5-fold cross-validation, meaning LM generates synthetic data independently 5 times using the same training data. The results at each ratio are based on these five independently generated datasets, significantly reducing randomness.
>
> Third, the effect of variance is further reduced by using 139 classifiers for each method and up to five resampling ratios per dataset (depending on the original imbalance). For each dataset, SM, LM, and LSH each have up to 695 candidate configurations (139 × up to 5 ratios). Because any remaining variance may occasionally favor LM or LSH, but each method is always represented by its best configuration, the selection process is almost variance-neutral.
>
> Fourth, evaluation across 60 datasets provides statistical coverage that suppresses incidental noise. Random fluctuations in LLM sampling cannot explain the systematic patterns observed across this many datasets.
>
> Finally, the empirical results contradict the idea that reduced variance explains LSH’s improvements. If fewer LLM calls made LSH more stable, LSH would outperform LM uniformly. Instead, LSH’s gains are observed primarily in highly imbalanced datasets, where the hybrid design is theoretically expected to help. In mid and less imbalanced datasets, SM, LM, and LSH are nearly identical.
>
> In summary, we think each method is evaluated exactly as it would be used in practice. The reduced number of LLM calls in LSH is not a methodological imbalance but a core aspect of the hybrid method’s design.
>
> ---
>
> ## **Weakness 2**
> > The performance gains are small (~1%), and Bayesian Sign Test (BST) shows many draws.
>
> **Response:**
> It is true that the average margin of LSH over LM and SM is modest when aggregated across all 60 datasets, and that the Bayesian Sign Test (BST) yields many draws. However, the strength of our work lies beyond global average margins; here are the details.
>
> First, the results from Section 5.1 (overall performance) were not our strength, and we mentioned this in the Discussion section, in line with prior work (Nuno Moniz and Hugo Monteiro. No free lunch in imbalanced learning. Knowledge-Based Systems, 227:107222, 2021.). The goal of this work is not to claim universal superiority but to evaluate how SM, LM, and LSH behave across diverse imbalance conditions. Given the diversity of the 60 datasets and the extensive classifier (with hyperparameter search), it is expected that no single oversampler will dominate across all datasets. The presence of many statistical draws is typical in such broad benchmarks.
>
> Second, the aggregate result masks the clear and substantial gains in the more imbalanced group. In the more imbalanced 20 dataset subset, LSH improves over SM by +0.049, and over LM by +0.025, with BST win probabilities of 0.88 (vs SM) and 0.46 (vs LM). These are not minor effects and directly reflect the hybrid design’s strength in settings where oversampling matters most.
>
> Lastly, efficiency is another core contribution. LSH reduces LLM generation cost with a single LLM call, transforming oversampling into a constant-time operation. Performance gains, even if modest on average, become meaningful when combined with significant efficiency improvements.
>
> While the average improvement across all datasets is small, this is expected given the diversity of benchmarks and extensive model search. What matters is that LSH shows substantial improvements under severe imbalance, a clear advantage in few-shot/zero-shot regimes, and major efficiency benefits. These combined contributions establish the value of the proposed hybrid oversampling strategy.

---

> ### Author Response · Authors · 2025-11-22
> **Authors' Response 2**
>
> ## **Weakness 3**
> > Only GPT-4o-mini is evaluated, limiting generalizability.
>
> **Response:**
> We acknowledge the reviewer’s concern that our experiments rely on a single LLM (GPT-4o-mini) and do not explore alternative models or decoding settings. It is reasonable to question whether this may introduce model-specific bias or limit generalizability. However, the strength of our method does not depend on the specific choice of LLM.
>
> First, the goal of this work is not to benchmark LLM architectures, but to introduce and evaluate a hybrid oversampling framework that combines an LLM for semantic expansion (quality) and SMOTE for scalable interpolation (efficiency). GPT-4o-mini serves as a representative high-quality model to demonstrate the viability of the hybrid design, not as the only possible choice. In the same context, we could consider other traditional oversampling methods besides SMOTE, but we chose SMOTE as a representative method. For sure, experimental results with more method combinations can provide more substantial evidence (you are definitely right). However, we think our choice (GPT-4o-mini + SMOTE) at least sufficiently demonstrates the feasibility of a hybrid method.
>
> Second, the core ideas that justify the hybrid approach are not model-specific. LLMs can extend the minority class distribution beyond available samples. SMOTE can then efficiently and cheaply interpolate. LLMs are slow when repeatedly called. LSH removes the repeated calls and achieves constant-time scaling. These are structural characteristics of any general-purpose LLM, not GPT-4o-mini in particular.
>
> Third, our results cover a highly diverse benchmark: 60 datasets × 139 classifiers × up to 5 ratios × 5 CV folds. A hybrid method relies on LLM seeds; therefore, different LLMs can yield different results. However, this diversity adopted in the experiments reduces the risk that the results depend on a specific LLM.
>
> In brief, while our experiments use a single LLM to demonstrate the feasibility of the hybrid approach, the LSH framework fundamentally does not depend on GPT-4o-mini’s specific behavior. The structural advantage of LSH, i.e., semantic expansion followed by efficient scaling, holds for any capable LLM. Exploring other LLMs is a natural next step, but it does not detract from the validity or generality of the proposed method.
>
> ---
>
> ## **Question 1**
> > Evaluate the LLM baseline using the same single-generation protocol as LSH.
>
> **Response:**
> We understand the reviewer’s suggestion to re-run the LM baseline using the same cumulative protocol as LSH (i.e., generate once at 1:0.2 and then scale up without repeated LLM calls). We believe this is closely related to the methodological concern raised earlier (Weakness 1). As explained in our response, our experimental design intentionally reflects how each method would be used in practice, not an artificially equalized internal workflow. We evaluate SM, LM, and LSH under their actual operational behavior, as a practitioner would. So, we think that forcing LM to behave like LSH would no longer be LM oversampling and would compromise the validity of the comparison.
>
> ---
>
> ## **Question 2**
> > Please analyze sensitivity to seed ratios and LLM parameters (model/temperature).
>
> **Response:**
> We agree that exploring different initial seed ratios, multiple LLM models, and varying decoding settings is a worthwhile direction to get insight into sensitivity and generalizability. However, these aspects do not directly support the core contribution of this work.
>
> First, as mentioned earlier, our goal in this study is to evaluate the proposed hybrid oversampling framework, not to exhaustively tune LLM parameters or benchmark many LLM families. The contribution is the design of LSH, using an LLM once to generate seeds and SMOTE to scale efficiently, not optimization of LLM hyperparameters.
>
> Second, exploring the initial seed ratio (e.g., 1:0.1, 1:0.2, etc.) is conceptually valuable, but it may not be possible in practice because a proper seed ratio can vary in each dataset and depends on the quality of generated seed samples. Even if we analyze sensitivity across all datasets, we may not obtain meaningful results. Also, the LSH mechanism is stable because SMOTE performs the majority of expansion, and the exact initial ratio has a limited influence. We already show robustness across extensive experimental settings. This demonstrates that the method’s behavior is not overly sensitive to a single design parameter.
>
> We, again, acknowledge that exploring seed ratios, multiple LLMs, and decoding settings is valuable, but these analyses fall outside the scope of this work. Our contribution is the hybrid oversampling mechanism itself, which is model-agnostic and already shown to be robust across extensive experimental settings. Sensitivity studies are the next step, but the absence of such tuning does not weaken the method's validity or generality.

---

### Author Response · Authors · 2025-11-22
**For all reviewers**

Recently, we read the ICML 2025 position paper by Kim et al., “The AI Conference Peer Review Crisis Demands Author Feedback and Reviewer Rewards,” which highlights growing concerns about declining review quality in major machine learning conferences. We have also heard similar concerns from colleagues who suspect that some recent reviews may have been generated by AI.

In contrast, the reviews we received for this submission are thorough, thoughtful, and highly constructive. We sincerely appreciate the time, effort, and expertise each reviewer invested. Your comments meaningfully contribute not only to improving our paper but also to strengthening the scientific dialogue in our community.

We still believe that our work can contribute to our research area, i.e., as a new oversampling approach to mitigate imbalance. So we want this rebuttal to be constructive, transparent, and active. If any of our responses are unclear, incomplete, or unconvincing, we welcome further questions or discussion. We genuinely appreciate your feedback and are committed to addressing all concerns to the best of our ability.

---

### Author Response · Authors · 2025-12-03
**Revised Paper is uploaded**

We uploaded a revised paper. Most of the reviewers' concerns have been resolved and explained in the revision.

To better understand our paper, we emphasize that the goal of this work is to validate the hybrid paradigm.

### First,
Although additional LLMs could be included, they are not necessary for our core contribution: the comparison must be between three roles: SMOTE (representative traditional oversampling), LLM (any LLM capable of generating seed samples), and LSH (our hybrid method using the two). If a different LLM were used, it would simply replace GPT-4o-mini in both LM and LSH, and the evaluation principle would remain unchanged; the purpose is to show that "a small amount of LLM + SMOTE" is more efficient than "LLM everywhere", not to pit LLMs against each other.

### Second,
Regarding the choice of seed ratio, our intention is to use the LLM as minimally as possible; hence, generating a small (e.g., 1:0.2) minority base and letting SMOTE scale afterward. Identifying an "optimal" seed ratio is not feasible because it depends on dataset characteristics, LLM behavior (or type), the classifier to be used, and the final target ratio; this variability makes the question itself an open research problem. We therefore set the seed ratio as the first target ratio (e.g., 1:0.2 or 1:0.4, depending on the original dataset class distribution) for all datasets as a practical and defensible choice.

### Third,
SOTA LLM-based tabular generators such as HARMONIC or LLMOverTab are not appropriate baselines for our study: they focus on maximizing generative quality using large models, multi-stage pipelines, or fine-tuning, and thus operate under computational regimes vastly different from ours. Our goal is to show that a lightweight hybrid strategy can substantially reduce LLM usage while maintaining competitive utility, not to outperform heavy SOTA generative systems.
These clarifications reinforce the validity of our design choices and align them with the purpose of the paper: demonstrating the practicality under imbalance and the efficiency of the LSH hybrid approach.

---

### Meta-Review · Area_Chair_AJMq · 2026-01-07

**Summary:**

The paper proposes a hybrid minority oversampling method that combines LLM generation with SMOTE, offering the efficiency of SMOTE with the strong presentation of LLMs. The reviewers acknowledge simplicity and practicality as a major strength of the paper. Also, a comprehensive evaluation enhances the contribution of the paper. However, the reviewers collectively point out following weaknesses:

- Limited performance improvement
- Limited baselines, comparison with SOTA or a fairly tuned baseline is required
- Limited exploration in LLM settings, including choice of models, decoding option, and sampling strategy
- Missing implementation details

Overall, the reviewers' opinion leans toward rejection. Therefore, I can't recommend acceptance for the paper.

**Reviewer Concerns:**

Resolved concerns
- Reviewer F81Y
  - Missing important details
- Reviewer paAB
  - Choice of seed ratio need to be justified

Remaining concerns
- Reviewer MEWy
  - Lack of competitive baselines with proper settings
  - Limited performance gain
  - Limited exploration in LLM setting
- Reviewer fuz8
  - Presentation needs improvement
  - Lack of justification for LLM uses
  - Insufficient qualitative example
  - Weak performance
  - Limited baselines, comparison with SOTA is recommended
- Reviewer F81Y
  - Limited exploration in LLM setting
- Reviewer paAB
  - Limited exploration in LLM setting
  - Insufficient analysis of dataset-specific difference

**Reviewer Scores:**

- Reviewer MEWy: Would maintain the current score.
- Reviewer fuz8: Would maintain the current score.
- Reviewer F81Y: Would maintain the current score.
- Reviewer paAB: Would maintain the current score.

---

### Decision · Program_Chairs · 2026-01-26

Reject